# Identification of a conserved S2 epitope present on spike proteins from all highly pathogenic coronaviruses

Rui P Silva[1†], Yimin Huang[1†], Annalee W Nguyen[2*†], Ching-Lin Hsieh[1], Oladimeji S Olaluwoye[3], Tamer S Kaoud[4], Rebecca E Wilen[2], Ahlam N Qerqez[2], Jun-Gyu Park[5,6], Ahmed M Khalil[5], Laura R Azouz[2], Kevin C Le[2], Amanda L Bohanon[1], Andrea M DiVenere[2], Yutong Liu[2], Alison G Lee[1], Dzifa A Amengor[1], Sophie R Shoemaker[7], Shawn M Costello[8], Eduardo A Padlan[9], Susan Marqusee[7,10], Luis Martinez-Sobrido[5], Kevin N Dalby[4], Sheena D'Arcy[3], Jason S McLellan[1,11*], Jennifer A Maynard[2,11*]

[1]Department of Molecular Biosciences, The University of Texas at Austin, Austin, United States; [2]Department of Chemical Engineering, The University of Texas at Austin, Austin, United States; [3]Department of Chemistry and Biochemistry, The University of Texas at Dallas, Dallas, United States; [4]Division of Chemical Biology and Medicinal Chemistry, The University of Texas at Austin, Austin, United States; [5]Texas Biomedical Research Institute, San Antonio, United States; [6]Laboratory of Veterinary Zoonosis, College of Veterinary Medicine, Chonnam National University, Gwangju, Republic of Korea; [7]Department of Molecular and Cell Biology, University of California, Berkeley, Berkeley, United States; [8]Biophysics Graduate Program, University of California, Berkeley, Berkeley, United States; [9]Retired, Kensington, United States; [10]Department of Chemistry, University of California, Berkeley, Berkeley, United States; [11]LaMontagne Center for Infectious Diseases, The University of Texas at Austin, Austin, United States

*For correspondence:
annalee@utexas.edu (AWN);
jmclellan@austin.utexas.edu
(JSMcL);
maynard@che.utexas.edu (JAM)

[†]These authors contributed equally to this work

**Abstract** To address the ongoing SARS-CoV-2 pandemic and prepare for future coronavirus outbreaks, understanding the protective potential of epitopes conserved across SARS-CoV-2 variants and coronavirus lineages is essential. We describe a highly conserved, conformational S2 domain epitope present only in the prefusion core of β-coronaviruses: SARS-CoV-2 S2 apex residues 980–1006 in the flexible hinge. Antibody RAY53 binds the native hinge in MERS-CoV and SARS-CoV-2 spikes on the surface of mammalian cells and mediates antibody-dependent cellular phagocytosis and cytotoxicity against SARS-CoV-2 spike in vitro. Hinge epitope mutations that ablate antibody binding compromise pseudovirus infectivity, but changes elsewhere that affect spike opening dynamics, including those found in Omicron BA.1, occlude the epitope and may evade pre-existing serum antibodies targeting the S2 core. This work defines a third class of S2 antibody while providing insights into the potency and limitations of S2 core epitope targeting.

## Editor's evaluation

This study presents valuable findings on the isolation of an antibody that recognizes a novel, highly conserved SARS-CoV-2 Spike (S) epitope, adding to the growing repertoire of anti-S antibodies cross-reactive against human and zoonotic coronaviruses. The authors provide solid evidence supporting their claims, although the proposed antiviral mechanism of the newly described antibody requires further validation, and in vivo effectiveness remains to be determined.

The work will be of interest to biologists working to develop pan-coronavirus vaccines and therapies.

## Introduction

The COVID-19 pandemic is the latest and largest of three deadly coronavirus outbreaks, including those caused by SARS-CoV in 2002 and MERS-CoV in 2012. Despite the successes of vaccines and antibody therapeutics that neutralize SARS-CoV-2 virus by disrupting interactions between the ACE2 receptor and the spike fusion protein, mutations accumulating primarily in the S1 domain have resulted in widespread evasion of antibodies elicited against early virus strains. This has culminated in the currently circulating Omicron subvariants with >30 amino acid changes that resist neutralization by all monoclonal antibodies with Emergency Use Authorization (*VanBlargan et al., 2022*; *Imai et al., 2023*) and cause breakthrough infections in fully vaccinated individuals. Moreover, the seven coronaviruses known to infect humans are closely related to strains found in wildlife, foreshadowing future coronavirus outbreaks.

The continued emergence of SARS-CoV-2 variants of concern underscores the need to identify therapeutic strategies more resistant to antigenic drift. Immunization with the entire spike ectodomain induces potently neutralizing antibodies that block binding of the receptor binding domain (RBD) to the ACE2 receptor (*Yuan et al., 2020*), indicating that the RBD is a protective and immunogenic, as well as variable, antigen. In contrast to the S1 domain comprising the RBD and N-terminal domains, the S2 domain is highly conserved, with 63–98% sequence similarity in pairwise comparisons across the seven human coronaviruses (*Figure 1—figure supplement 1*). Early in the pandemic, S2-directed antibodies often dominated the immune repertoire in convalescent patients, indicating that at least some S2 epitopes are immunogenic (*Voss et al., 2021*). Moreover, the functionally analogous domains in the fusion proteins from influenza virus, respiratory syncytial virus, and human immunodeficiency virus are targeted by protective antibodies (*Impagliazzo et al., 2015*; *Corti et al., 2017*), supporting the hypothesis that the spike S2 domain may also be an effective target.

Whereas antibodies binding the spike RBD have been rigorously classified based on epitope recognized (*Barnes et al., 2020*) and this information used to support development of RBD mosaic vaccines (*Cohen et al., 2022*), a complementary analysis of the S2 domain is in its infancy. Fewer than 5% of the ~7000 anti-SARS-CoV-2 spike monoclonal antibody sequences in the CoV-AbDab database bind S2 as of July 2022 (*Raybould et al., 2021*). Moreover, just two classes of S2 binding antibodies have been described in the literature: those binding the fusion peptide and adjacent S2' cleavage site (*Dacon et al., 2022*; *Low et al., 2022*; *Sun et al., 2022*) and those binding the S2 stem proximal to the viral membrane (*Hsieh et al., 2021*; *Pinto et al., 2021*; *Sauer et al., 2021*; *Zhou et al., 2022*). Here, we define a third class of S2 antibody that binds the highly conserved S2 hinge region, which converts from a bent hairpin to extended alpha helix during the pre-to-post-fusion spike conformational change. Evaluation of the S2 hinge epitope and the impact of spike dynamics on epitope accessibility will inform our understanding of the role of S2 domain epitopes in antibody recognition.

## Results

### A MERS S2 mouse immune library yields antibody 3A3 that also binds SARS-CoV-2 spike

Balb/c mice were immunized with stabilized MERS-CoV S2 protein and boosted 4 weeks later, resulting in robust serum antibody titers against the immunogen detectable at >1:10,000 dilution. The MERS-CoV S2 protein MERS SS.V1 spans residues 763–1291 of the MERS-CoV spike protein with a C-terminal T4 phage fibritin (foldon) domain that assembles into a prefusion trimer (*Hsieh et al., 2021*). An immune antibody library with ~3.1 × $10^8$ individual clones expressed as scFv-c-myc tag-pIII fusion proteins was generated and displayed on M13 bacteriophage. After 3–4 panning rounds, >80 clones binding both MERS S2 and SARS-2 spike were characterized, with ~85% of clones binding the shared foldon domain by ELISA (*Figure 1—figure supplement 2*). One foldon binder, 3E11, was carried forward as a control antibody along with the most promising spike-binding clone identified by phage ELISA, 3A3.

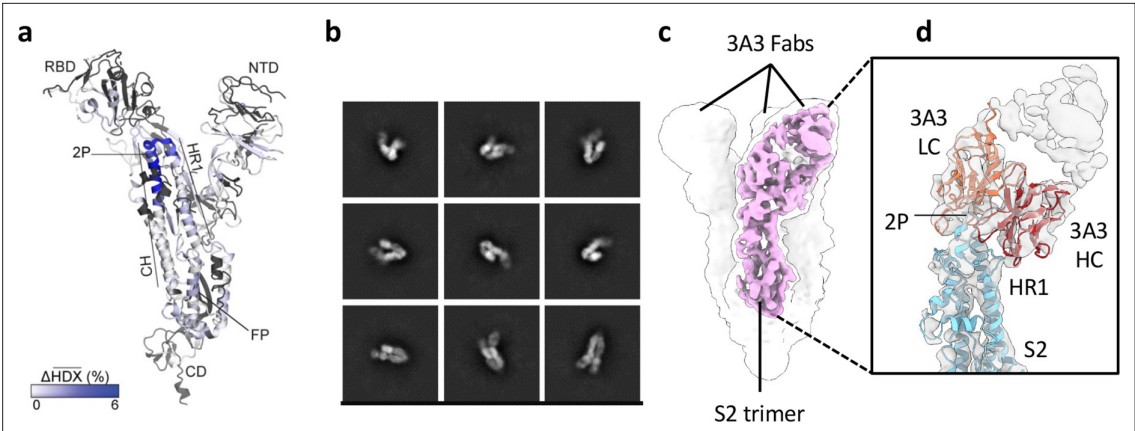

**Figure 1.** The hinge epitope spans the HR1/CH helices at the S2 apex. (**a**) Monomeric SARS-2 2P spike (PDBID: 6VSB chain B) colored according to the HDX difference in deuterium fractional uptake between SARS-2 HexaPro spike alone and with 3A3 IgG at $10^2$ s exchange. Residues lacking coverage are indicated in black. The figure was prepared using DynamX per residue output without statistics and PyMOL. (**b**) Cryo-EM 2D class averages of the SARS-2 S2 subunit bound to the 3A3 Fab. (**c**) Cryo-EM 3D reconstruction of the S2–3A3 Fab complex showing each S2 protomer binding one 3A3 Fab molecule. The pink 3D volume was generated from a focused refinement of one S2 protomer and 3A3 Fab. (**d**) A structure of the S2 subunit and a predicted structure of the 3A3 Fab shown as ribbons and fit into the cryo-EM map. The 3A3 Fab light (LC, orange) and heavy (HC, red) chains sandwich the apex of the spike S2 hinge (cyan).

The online version of this article includes the following source data and figure supplement(s) for figure 1:

**Source data 1.** HDX summary table for spike peptides.

**Figure supplement 1.** Sequence conservation is higher in the S2 domain than the S1 domain across coronaviruses that infect humans.

**Figure supplement 2.** Many cross-reactive scFv-phage target the foldon domain.

**Figure supplement 3.** Peptides monitored through all timepoints of deuteration for SARS-2 HexaPro spike alone and with 3A3 IgG or Fab.

**Figure supplement 4.** Deuterium uptake of SARS-2 HexaPro spike alone suggests the trimer is maintained under HDX conditions.

**Figure supplement 5.** HDX identified the apex of the S2 domain as the 3A3 epitope.

**Figure supplement 6.** HDX identified the spike-binding paratope in 3A3 IgG.

## Antibody 3A3 binds the S2 core at the HR1/CH hairpin hinge

Antibody 3A3 binds a highly conserved, conformational epitope spanning residues 980–1006 of the SARS-2 spike, at the apex of the S2 domain, distal to the viral envelope (*Figure 1a*), as determined by a combination of hydrogen-deuterium exchange mass spectrometry (HDX) and low-resolution cryo-EM. This region spans the hairpin hinge, joining the heptad repeat 1 (HR1) and the central helices (CH), and is referred to hereafter as the hinge epitope. This region plays a critical role in the spike conformational changes required for fusion of the viral envelope and target cell membrane. In the intact spike homotrimer, the membrane-proximal, stalk-like S2 domain is capped by S1 whose N-terminal and receptor-binding domains (RBD) form a responsive surface allowing each RBD to extend to an 'up' position for receptor binding or tuck into a 'down' position for immune shielding of the receptor-binding site. After the RBDs engage a receptor in the up position and target cell proteases prime the spike, the S1 domain is released from S2, propelling the fusion peptide into the target cell or endosomal membrane. The hinge then extends to form an alpha helix that bridges the viral envelope and target cell membrane, initiating fusion and leaving the spike in the post-fusion state (*Cai et al., 2020*).

For HDX epitope analysis, we measured deuterium uptake of the SARS-2 HexaPro (*Hsieh et al., 2020*) spike alone and when bound by the 3A3 IgG or 3A3 Fab (*Supplementary files 1 and 2*). We tracked 192 unmodified peptides (*Figure 1—figure supplement 3*) through the deuteration time course ($10^1$, $10^2$, $10^3$, and $10^4$ s). Analysis of the raw deuterium uptake in the SARS-2 HexaPro spike alone is consistent with a trimer during exchange with relatively low deuterium uptake in the helix at the center of the trimer and high deuterium uptake in the HR1 helix at the trimer's surface (*Figure 1—figure supplement 4a*). Analysis of the isotopic distribution width of peptides from regions of spike reported to display bimodal spectra (*Costello et al., 2022*) further suggests conformational heterogeneity consistent with the trimeric spike sampling an open conformation (*Supplementary file 3* and

*Figure 1—figure supplement 4b*). Antibody epitopes were identified by examining the difference in deuterium uptake between SARS-2 HexaPro spike in the free and antibody-bound states (*Figure 1a*). We defined a significant difference as greater than 0.2 Da with a p-value <0.01 (*Figure 1—figure supplement 5a and b*). The binding of 3A3 IgG caused a significant decrease in 12 peptides that redundantly span residues 980–1006 of the SARS-2 HexaPro spike (*Figure 1—figure supplement 5c*). These peptides have reduced deuterium uptake with 3A3 IgG at several timepoints during the exchange reaction. A similar result was obtained with the 3A3 Fab (*Figure 1—figure supplement 5c and d*).

Antibody paratopes were similarly identified by comparing deuterium uptake of 3A3 IgG alone to that with an excess of SARS-2 HexaPro spike. We monitored 169 peptides that redundantly cover 80% of the 3A3 IgG sequence. This analysis implicated CDRs L2 (residues 53–59) and H3 (residues 105–109) as forming the paratope that interacts with SARS-2 HexaPro spike (*Figure 1—figure supplement 6*). The spike epitope identified by HDX is consistent with low-resolution cryogenic electron microscopy (cryo-EM) of 3A3 Fab bound to stabilized SARS-2 S2, which shows Fabs bound to the apex of each S2 protomer in a 1:1 stoichiometry (*Figure 1b–d*). The open S2 conformation of spike resulted in particles with varying degrees of protomer opening that precluded sorting into 3D classes for high-affinity structural resolution.

## Access to the hinge epitope depends on spike domain dynamics

Mapping of the hinge epitope onto full-length spike structures shows that this region is fully occluded by the S1 domain in the closed (three RBDs down) state but becomes increasingly visible in structures with one, two, or three RBDs up and with ACE2 bound (*Figure 2a*). In fact, RBDs in the down position make direct hydrogen bonds with the 3A3 epitope at residues 983–988 (*Hossen et al., 2022*), thereby excluding other binding interactions. Consistent with this structural analysis, simultaneous binding of ACE2 and 3A3 to spike was demonstrated by BLI in which immobilized 3A3 captured SARS-2 HexaPro and then soluble ACE2. In a similar experiment, control mAb 2–4, whose epitope spans adjacent RBDs in the down state, bound SARS-2 HexaPro but could not then bind ACE2 (*Figure 2b*). Additionally, 3A3 did not bind SARS-2 HexaPro spike locked into the closed conformation by engineered disulfide bonds (*Henderson et al., 2020*), although this constrained spike was recognized by mAb 2–4 (*Figure 2c*). As expected from the structural analysis, 3A3 binding does not block ACE2 binding to the spike RBDs. Moreover, the hinge epitope is only accessible when RBDs have freedom to convert to the up position.

Access to S2 core epitopes is only partially understood, as are the dynamics of spike breathing and other complex intra-protein spike motions. *Costello et al., 2022* showed that stabilized spike undergoes reversible protomer opening in solution to expose the S2 core and the hinge epitope. They performed an independent HDX experiment under conditions favoring the open trimer conformation to show that 3A3 exclusively binds and stabilizes an S2-open state, distinct from the open/closed states used to describe RBD motion. Consistent with this report, 3A3 has a faster on-rate for spike variants favoring S2 opening versus unmodified spike. SARS-2 HexaPro bearing an E1031R substitution, which disrupts an electrostatic interaction between E1031 and R1039 on adjacent protomers deep in the S2 base (*Figure 2d*), favored the S2-open state relative to unmodified HexaPro as assessed by HDX (*Figure 2—figure supplement 1*) and exhibited a fourfold increased on-rate for 3A3 binding ($2.9 \pm 0.1\ \mu M^{-1}\ s^{-1}$) versus unmodified HexaPro spike ($0.8 \pm 0.1\ \mu M^{-1}\ s^{-1}$; *Figure 2e*). This indicates that 3A3 binding to full-length spike occurs after the RBDs have transitioned to the up position and the S2 domain has relaxed into a more open state. Overall 3A3 binding rates thus depend on these transition rates in addition to typical antibody-antigen association and dissociation rates (*Figure 2f*).

## Antibody 3A3 binds a conformational epitope spanning the 2P stabilizing mutations

To validate the HDX and cryo-EM data and define the 3A3 epitope with single amino acid resolution, 16 solvent-exposed epitope residues were individually altered in HexaPro to assess the impact on 3A3 binding. Three changes (L984A, Q992L, and R995A) improved 3A3 binding, while five (D985L, E988Q or I, D994A, and L1001A) significantly reduced 3A3 binding by ELISA (*Figure 3a*, *Figure 3—figure supplement 1*). Substitutions nearly ablated binding at positions D985 and E988, which form a negatively charged patch adjacent to the stabilizing 2P changes, P986 and P987 (*Figure 3b and c*). Since

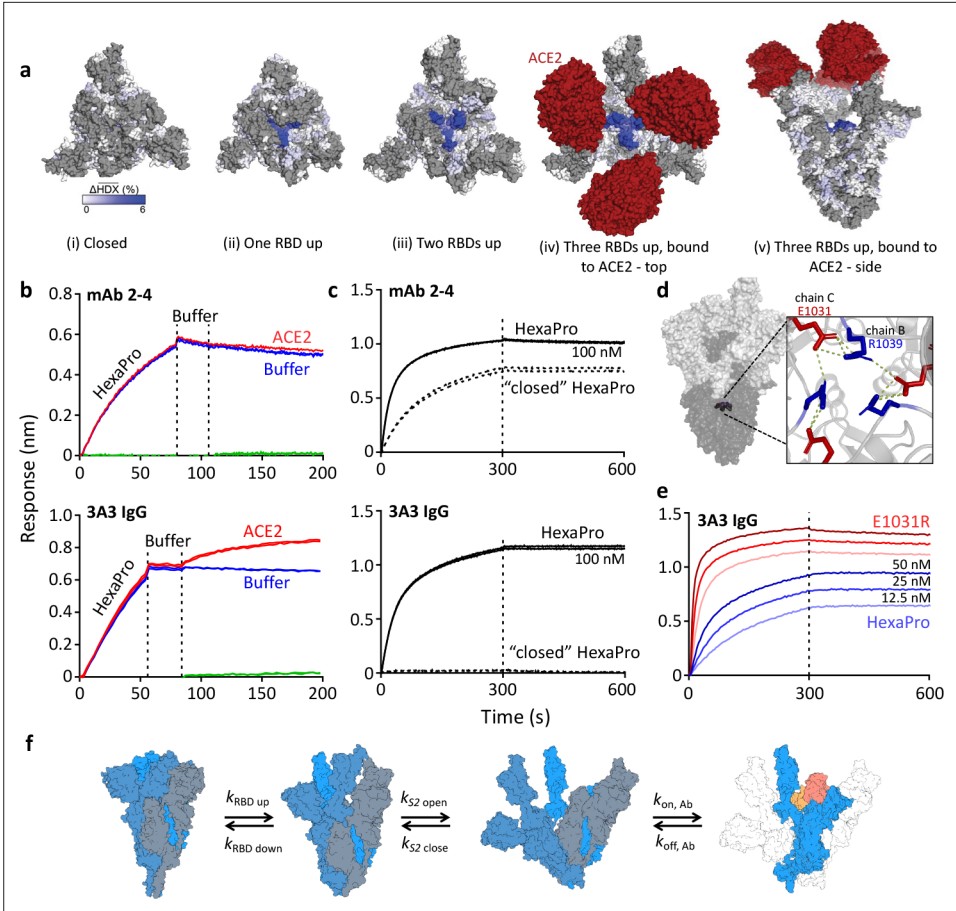

**Figure 2.** The hinge epitope is accessible only in an RBD-up and S2-open spike conformation. (**a**) Trimeric SARS-2 spike in various conformations colored according to difference in deuterium fractional uptake between SARS-2 HexaPro spike alone and with 3A3 IgG. The hinge epitope within S2 is colored dark blue in structures of wild-type SARS-2 spike in the (i) three RBDs down or closed conformation (PDB: 6XR8) and in structures of stabilized spike with (ii) one RBD up (PDB: 6VSB), (iii) two RBDs up (PDB: 7A93), or (iv) three RBDs up while bound to ACE2 (red) in top-view and (v) sideview (PDB: 7A98). Residues lacking coverage in the HDX experiment are indicated in gray. (**b**) Antibody 3A3 (bottom) or control mAb 2–4 (top) were coupled to anti-Fab BLI sensors and allowed to capture HexaPro or nothing (buffer, green line), then dipped into buffer (baseline), and finally dipped into ACE2-Fc (ACE2, red) or nothing (buffer, blue). (**c**) BLI binding of immobilized control mAb 2–4 (top) or antibody 3A3 (bottom) to 100 nM HexaPro (solid) or HexaPro locked into the 'closed' conformation (dashed). Vertical dashed lines indicate start of dissociation phase. (**d**) The network of hydrogen bonds formed by residues E1031 and R1039 across protomers deep in the S2 core is shown on intact HexaPro spike and in detail in a top view (PDB: 6XKL). (**e**) Antibody 3A3 was coupled to anti-Fc BLI sensors and allowed to bind HexaPro or E1031R HexaPro (E1031R) spike protein. All BLI data are representative of biological duplicates. Each experiment was repeated in technical duplicate except e, which was tested once at each concentration to allow all data to be collected simultaneously for direct comparison. (**f**) Model of the kinetic changes required for antibody binding to the hinge epitope, including conversion of the RBDs into the up position and some degree of opening of the S2 domain in addition to typical antibody association and dissociation kinetics (generated using PDB 6XV8 and 7A98).

The online version of this article includes the following source data and figure supplement(s) for figure 2:

**Source data 1.** BLI sensorgram data.

**Figure supplement 1.** HexaPro E1031R variant exposes the hinge epitope.

E988Q is present in the spike proteins of α-coronaviruses NL63 and 229E, this suggests 3A3 binding may be limited to β-coronavirus spike proteins. When SARS-2 D614G lentivirus containing D985L, E988A, or E988Q substitutions were evaluated for the ability to infect ACE2-expressing HEK 293 cells, all had impaired infectivity (55–98% reduction at the highest titer tested; *Figure 3d*), suggesting that escape mutations within this epitope have a high fitness cost. Indeed, GISAID genomic sequence data

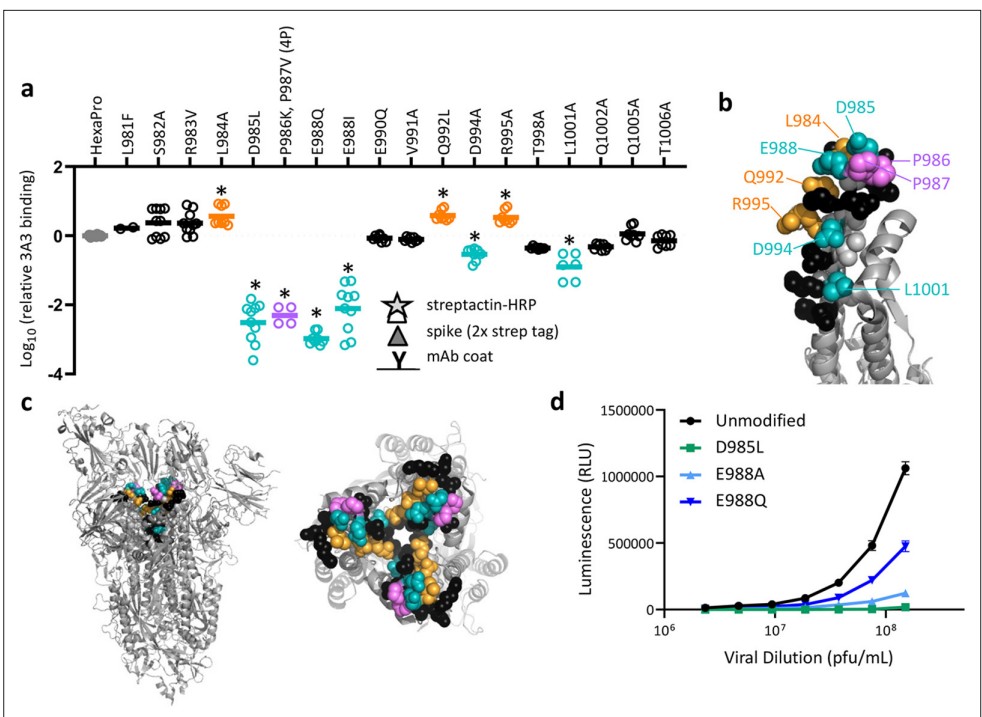

**Figure 3.** SARS-2 spike residues 985–988 are recognized by 3A3 and impair spike function upon substitution. (**a**) Residues important for 3A3 binding were identified by single residue changes in HexaPro that increased or decreased binding to 3A3 relative to HexaPro. Each variant was tested with duplicate technical replicates in 2–6 independent ELISA assays. Significance relative to unaltered HexaPro was determined by ANOVA with post-hoc Tukey–Kramer test with $\alpha$ = 0.01; data meeting this criterion indicated by *. (**b**) Location of the residue changes altering binding to 3A3 shown in the HexaPro spike (6XKL) monomer and in (**c**) intact spike (side view) and the S2 domain (top-down view). All epitope residues (980–1006) are shown in space-fill, with residues colored according to their effect on 3A3 binding: improved binding (orange), reduced binding (teal), no effect (black), and those not altered (gray). The 2P stabilizing mutations within the hinge epitope are displayed in purple. (**d**) The infectivity of lentivirus pseudotyped with unmodified D614G SARS-2 spike or variants with D985L (green), E988A (light blue), and E988Q (dark blue) substitutions was compared by luciferase activity. Data shown are the mean luminescence with standard deviation of three technical replicates.

The online version of this article includes the following source data and figure supplement(s) for figure 3:

**Source data 1.** ELISA binding data and relative luminescence data for pseudovirus infection assays.

**Figure supplement 1.** HexaPro variants with reduced 3A3 binding retain trimer SEC profile.

**Figure supplement 2.** The hinge epitope is nonlinear and inaccessible in aggregated or misfolded protein.

**Figure supplement 3.** Original blot images for *Figure 3—figure supplement 2a*.

**Figure supplement 4.** Adding a disulfide bond slightly improves 3A3 binding by ELISA to spike with P986 and P987 reverted to the native sequence.

analyzed by Los Alamos National Laboratory's COVID-19 Viral Genome Analysis Pipeline (*Korber et al., 2020*) accessed in August 2022 showed that the Shannon entropy of positions 985 and 988 was >40-fold and >150-fold reduced relative to the average Shannon entropy of the SARS-2 S2 domain and full SARS-2 spike ectodomain, respectively. Pseudotyped virions bearing spike with 988Q found in $\alpha$-coronaviruses were least impacted, consistent with tolerance to this substitution for spike function. Importantly, single mutations within this epitope that have emerged, L981F in SARS-2 Omicron BA.1 spike (which reverted in subsequent Omicron variants BA.2 through BA.5) and S982A in SARS-2 Alpha spike, had no significant impact on 3A3 binding (*Figure 3a*).

Antibody 3A3 could not detect fully denatured coronavirus spike in Western blot (*Figure 3—figure supplement 2a*, *Figure 3—figure supplement 3*), consistent with recognition of the folded, bent S2 hinge conformation. After three freeze/thaw cycles, aggregates were detectable by SDS-PAGE in the SARS-2 spike but not in the stress-resistant (*Hsieh et al., 2020*) SARS-2 HexaPro. (*Figure 3—figure*

supplement 2b). The control foldon-binding antibody 3E11 bound fresh and stressed proteins similarly, but 3A3 binding to stressed 2P spike was greatly decreased (~150-fold worse ELISA $EC_{50}$), while binding of stressed SARS-2 HexaPro was unaffected (*Figure 3—figure supplement 2c and d*). These data are consistent with the HDX and cryo-EM data indicating that 3A3 binds properly folded, prefusion spike.

Residues K986 and V987 are substituted with prolines in the stabilized SARS-2 2P and HexaPro soluble spikes used by most laboratories as these changes substantially improve the yield and stability of soluble prefusion spike. Given their proximity to hot spot residues D985 and E988 within the hinge epitope, we reverted 2P to the native sequence and evaluated the impact on 3A3 binding (*Figure 3a–c*). Binding to this 4P spike (HexaPro with P986K and P987V) was dramatically impaired. Since proline side chains are part of the main chain backbone, the 2P changes may serve to rigidify the epitope and/or the native lysine residue may introduce steric or electrostatic clashes. To mimic the prefusion bent conformation without 2P, we introduced a disulfide bond between amino acids 965 and 1003 (*Hsieh et al., 2020*) in 4P to create 4P-DS. By ELISA, 3A3 binding to 4P-DS was partially recovered (*Figure 3—figure supplement 4*). Collectively, these data demonstrate that antibody 3A3 binds a conformational hinge epitope dominated by residues D985 and E988 and rigidified by the adjacent stabilizing 2P changes.

## Antibody RAY53 binds the authentic SARS-2 spike hinge epitope

To understand the role of the hinge epitope in the context of authentic spike, we expected that an engineered version of 3A3 would be necessary to accommodate 986K and thereby improve binding to 4P, 4P-DS, and authentic spike. We evaluated several humanized 3A3 variants as previously described (*Nguyen et al., 2015*), yielding hu3A3 which bound HexaPro similarly to 3A3 by ELISA (*Figure 4—figure supplement 1*). To identify variants binding 4P-DS more strongly, two hu3A3 Fab libraries of ~3 × 10^7 members each were generated in a yeast display plasmid: a random mutagenesis library (*Fromant et al., 1995*) with an error rate of 0.3% and a site-directed mutagenesis library targeting three residues in CDRL2 and five residues in CDRH3 implicated in spike recognition by HDX. After 3–4 rounds of sorting for enhanced 4P-DS binding (*Figure 4—figure supplement 2*), individual clones were isolated. Combinatorial expression of selected $V_H$ and $V_L$ regions as IgG1 antibodies followed by ELISA screening for 4P-DS binding identified RAY53, comprised of a light chain from the site-directed library and a heavy chain from the random mutagenesis library.

RAY53 shows greatly improved binding to 4P-DS spike versus 3A3 (*Figure 4—figure supplement 3a and b*) while retaining 3A3's epitope sensitivity (*Figure 4—figure supplement 3c*). Fab 3A3 binds stabilized SARS-2 HexaPro S2 with ~3 nM equilibrium $K_d$, as measured by BLI and SPR (*Figure 4—figure supplement 4a and b*); RAY53 binds SARS-2 HexaPro similarly (*Figure 4—figure supplement 4f*). However, while 3A3 IgG1 binding to 4P-DS and 4P spikes was too weak for quantitation at the concentrations used (*Figure 4—figure supplement 3*), SPR analysis of RAY53 IgG1 binding to 4P-DS indicated a $K_d$ of 100 ± 16 nM (*Figure 4—figure supplement 4c*). Interestingly, the RAY53 Fab $K_d$ was 1.3 ± 0.2 μM, ~13-fold lower than the corresponding IgG (*Figure 4—figure supplement 4d*), suggesting both IgG arms simultaneously engage protomers within the same spike in this SPR orientation, consistent with cryo-EM images (*Figure 1b*). Overall, RAY53 retained binding to stabilized HexaPro spike while accommodating the native K986/V987 hinge residues.

## The hinge epitope is highly conserved across β-coronaviruses but susceptible to structural occlusion

The spike hinge at SARS-2 amino acids 980–1006 exhibits high sequence and structural conservation across all β-coronaviruses known to infect humans (*Figure 4a and b*), with Cα atom RMSDs ranging from 0.6 Å for HKU1 to 3.1 Å for MERS. To assess the phylogenetic range of spikes recognized by 3A3 and RAY53, binding to diverse coronavirus spikes was assessed by ELISA (*Figure 4c*). Antibodies 3A3 and RAY53 bound each of the 2P stabilized spikes similarly, with improved RAY53 binding observed for SARS-2 4P, 4P-DS, and HKU1. Binding of 3A3 to proline-stabilized SARS-2 HexaPro, SARS-2, aglycosylated SARS-2 HexaPro, SARS-1 and MERS spike was apparent with BLI-measured on-rates of ~0.2–1.3 μM^−1 s^−1; equivalent association rates were observed for RAY53 binding to the stabilized spikes tested (*Figure 4—figure supplement 4f*). Although both 3A3 and RAY53 bound SARS-2 HexaPro Omicron BA.1 by BLI, the on-rate was reduced ~15-fold relative to HexaPro. Binding

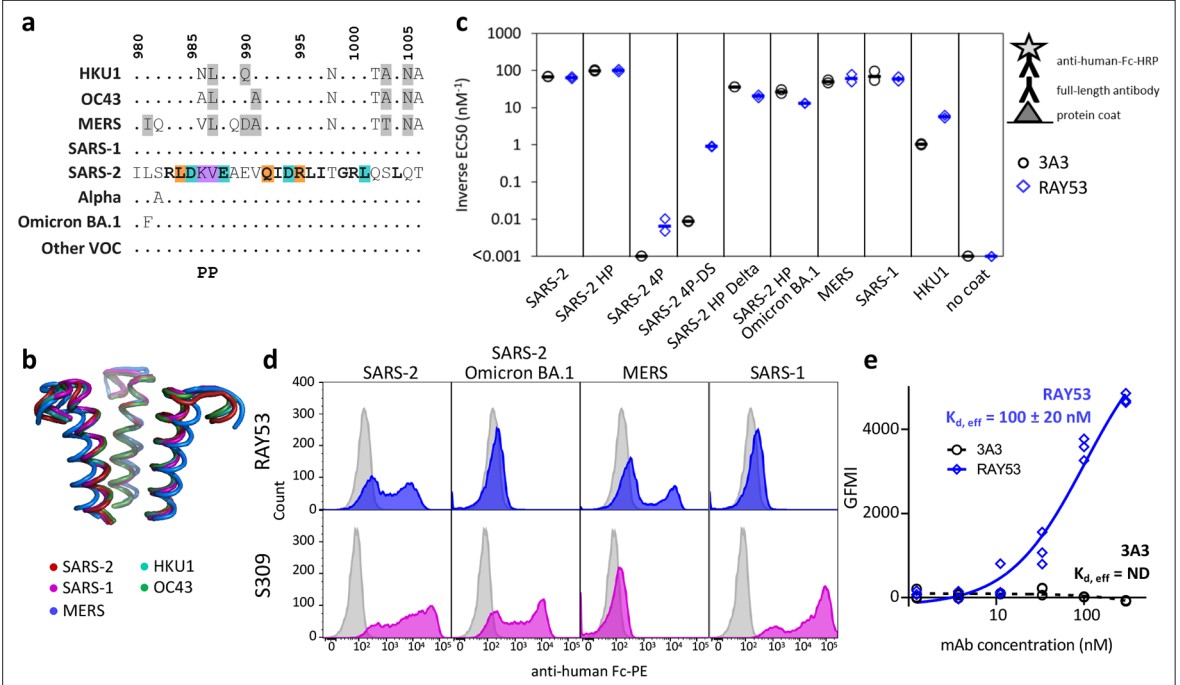

**Figure 4.** The hinge epitope is conserved across β-coronaviruses and variably accessible in authentic spike. The hinge epitope recognized by 3A3 (SARS-2 amino acids 980–1006) is highly conserved across the spike (**a**) sequences and (**b**) structures of β-coronaviruses known to infect humans, including Alpha, Omicron BA.1, and other variants of concern (VOC; Beta, Gamma, Delta, Epsilon, Omicron BA.2 through BA.5). In (**a**), identical residues are indicated by a dot and similar residues are highlighted in gray. Residues conserved across all listed β-coronaviruses are in bold. Residues that lost binding to 3A3 when altered as shown in *Figure 3* are in teal highlight and those whose disruption improved binding are orange. The location of the two proline mutations introduced to 2P variants are shown below the alignment. In (**b**), the structure of each epitope is displayed as follows: SARS-2 (6VSB) – red, SARS-1 (6CRV, RMSD = 0.8 Å) – magenta, MERS (5X5C, RMSD = 3.1 Å) – blue, HKU1 (5I08, RMSD = 0.5 Å) – teal, OC43 (6OHW, RMSD = 0.6 Å) – green. (**c**) Binding of full-length antibody 3A3 (black circles) and RAY53 (blue diamonds) to ancestral SARS-2, SARS-2 HexaPro (SARS-2 HP), SARS-2 4P, SARS-2 4P-DS, SARS-2 HexaPro Delta (SARS-2 HP Delta), SARS-2 HexaPro Omicron BA.1 (SARS-2 HP Omicron BA.1), MERS, SARS-1, HKU1, or milk (no coat) proteins by ELISA. Data are representative of duplicate biological replicates, each with duplicate technical replicates. The data midpoint is indicated with a bar. (**d**) Plasmids encoding full-length unstabilized spike proteins from SARS-2, SARS-2 Omicron BA.1, MERS, or SARS-1 were transiently transfected to Expi293 cells. The spike (blue or magenta histograms) or mock (grey histograms) transfected cells were stained with 100 nM RAY53 (top panels) or 10 nM control antibody S309 (bottom panels), followed by goat-anti-human Fc-PE secondary antibody, and flow cytometry scanning of 10,000 cells. The data shown are representative of triplicate experiments, with each condition repeated in technical duplicate. (**e**) Expi293 cells were transiently transfected with plasmids encoding SARS-2 spike and EGFP or EGFP only, then incubated with 3A3 (black circles) or RAY53 (blue diamonds) antibody (~1–300 nM) and anti-human Fc-PE before flow cytometric determination of the geometric mean fluorescence intensity (GMFI) in the PE channel for all green fluorescent cells. The GMFI of cells transfected with EGFP only was subtracted from the GMFI of cells expressing spike at each concentration, and the data fit to a three-parameter logistic curve to determine the effective $K_d$ ($K_{d,eff}$) for antibody binding. The data shown are representative of triplicate experiments; ND, not detected.

The online version of this article includes the following source data and figure supplement(s) for figure 4:

**Source data 1.** ELISA data and flow cytometry mean fluorescence intensity data.

**Figure supplement 1.** Antibody hu3A3 binds SARS-2 HexaPro similarly to 3A3 by ELISA.

**Figure supplement 2.** Antibody hu3A3 yeast display libraries were enriched for binding to 4P-DS.

**Figure supplement 3.** Antibody RAY53 retains epitope specificity while exhibiting higher affinity than 3A3 for SARS-2 HexaPro spike with reverted 2P changes.

**Figure supplement 4.** Antibodies 3A3 and RAY53 have low-to-mid nanomolar affinities for stabilized SARS-2 spike variants.

of RAY53 was maintained for diverse β-coronavirus spike proteins, consistent with the high sequence identity of this epitope.

To evaluate the binding of RAY53 to a range of unmodified β-coronavirus spikes, authentic SARS-2 (wild-type and Omicron BA.1), MERS, and SARS-1 spikes were displayed on the surface of Expi293 cells and stained with antibody (*Figure 4d*). The soluble expression of authentic coronavirus spike without stabilizing mutations results in poor yield and misfolded proteins, preventing accurate measurement

of binding affinities. Moreover, measurement of an 'effective' affinity based on binding to many spike proteins on the mammalian cell surface (*Feldhaus and Siegel, 2004*) is more indicative of antibody interactions with authentic spike during infection. Excluding the non-binding population consequent to transient expression, control antibody S309 bound SARS-2 and SARS-1 spike-expressing cells but not MERS spike-expressing cells, as expected (*Figure 4d*). Binding of RAY53 to wild-type SARS-2 spike yielded an effective $K_d$ of 100 ± 20 nM, nearly identical to the 100 nM affinity measured by SPR for 4P-DS (*Figure 4—figure supplement 4c*), with no 3A3 binding above background detected (*Figure 4e*). Overall, we conclude that affinity maturation to 4P-DS improved RAY53 binding to the hinge epitope found in authentic spike.

Interestingly, RAY53 bound SARS-2 and MERS spikes, but not SARS-2 Omicron BA.1 or SARS-1 spikes. The contrast between high RAY53 binding to stabilized SARS-2 HexaPro Omicron BA.1 (*Figure 4c*) and dramatic loss of binding to cell-surface displayed Omicron BA.1 implicates structural differences between the two spike formats as opposed to the single epitope mutation which is inert on its own (L981F, *Figure 4—figure supplement 3c*). SARS-2 Omicron BA.1 spike has accumulated mutations that result in tight packing of the RBDs in the down state, occluding many neutralizing RBD epitopes and aiding in immune evasion (*Gobeil et al., 2022*). We expect this tightly packed RBD surface will also impede access to the S2 core, including the hinge epitope. Similarly, strong RAY53 binding to stabilized SARS-1 spike was completely lost when spike was expressed without stabilizing mutations on the cell surface, despite an unaltered hinge epitope (*Figure 4a*). Overall, these data suggest mutations distal to the hinge epitope can restrict antibody access and this effect is lost in proline-stabilized spike.

## Targeting the hinge epitope inhibits cellular fusion and neutralizes pseudovirus but not authentic virus infection and mediates Fc effector functions

To investigate the impact of antibody binding on hinge function, we first employed a mammalian cell fusion assay (*Figure 5—figure supplement 1*). A CHO cell line expressing wild-type SARS-2 spike and EGFP was incubated with ACE2-expressing HEK 293 cells stained with red fluorescent Cell Trace Far Red. After 24 hr, large syncytia formed with green CHO cell fluorescence overlapping ~70% of red HEK 293 cell fluorescence, indicating fusion of the CHO and HEK 293 membranes in the presence of no antibody or 670 nM irrelevant human IgG1. Incubation with 670 nM or 67 nM of 3A3 significantly reduced colocalization to ~50% (p<0.0001) and significantly reduced syncytia size was noted down to 6.7 nM. These data indicate that 3A3 can prevent spike's ability to fuse viral and cell membranes.

We next compared 3A3 and RAY53 in an in vitro pseudovirus neutralization assay to determine whether stronger epitope binding improved neutralization. These antibodies, the potently neutralizing antibody S309 (*Pinto et al., 2020*), or an isotype control antibody were incubated with lentivirus expressing authentic spikes and added to HEK293 cells expressing the relevant receptor with infection monitored by luciferase expression (*Figure 5a*). Control antibody S309 potently neutralized SARS-1 and SARS-2 spike lentivirus ($IC_{50}$ of ~0.5 nM), similar to MLV pseudovirus reports (*Pinto et al., 2020*), but was approximately tenfold less potent against SARS-2 Omicron BA.1 and ineffective against MERS pseudoviruses, as expected (*Cameroni et al., 2022*). By contrast, 3A3 and RAY53 weakly and incompletely blocked infection of wild-type SARS-2 and MERS pseudoviruses (estimated $IC_{50}$ values >50 nM) but did not block infection by SARS-1 or SARS-2 Omicron BA.1 pseudoviruses. Incomplete pseudovirus neutralization has been noted in other studies, particularly with antibodies that do not directly block ACE2 binding, although it is unclear why this occurs (*Rogers et al., 2020*). Despite greatly improved binding to 4P-DS (*Figure 4—figure supplement 3a and b*) and authentic SARS-2 spike on the mammalian cell surface (*Figure 4e*), RAY53 did not neutralize pseudovirus better than 3A3, indicating that antibody binding is not the rate-limiting step in neutralizing the hinge epitope (*Figure 2f*).

With the understanding that these considerations may be different still in a live coronavirus context, neutralization of authentic SARS-2 wild-type virus by 3A3 and RAY53 alongside an isotype control and S309 positive control was tested in vitro (*Figure 5b*). In contrast to pseudotyped lentivirus, neither 3A3 nor RAY53 exhibited neutralization of the authentic virus in this assay. This difference may be caused by several factors, including the specific neutralization protocol used or variation in spike structure and/or dynamics in the specific environment of the virion surface.

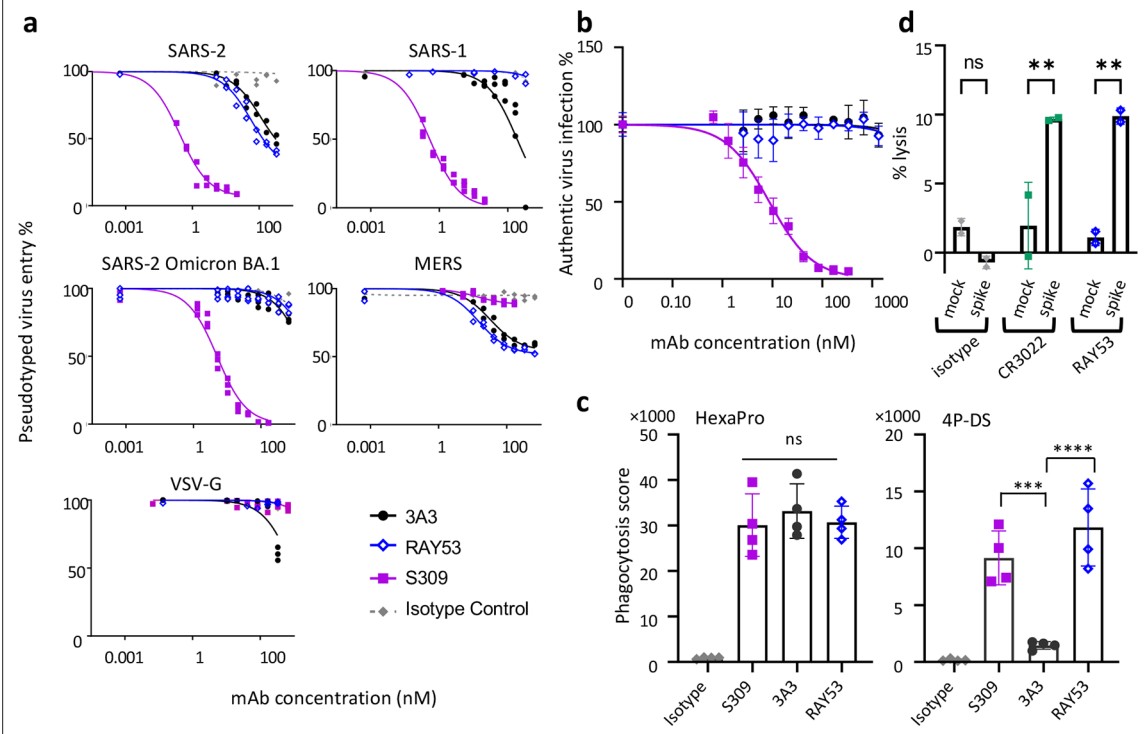

**Figure 5.** Targeting the hinge epitope recruits Fc effector functions. (**a**) Neutralization was evaluated by pre-incubating antibody with pseudotyped HIV particles that were then added to HEK 293T cells stably expressing ACE2 (SARS-1 and SARS-2 pseudoviruses) or DPP4 (MERS pseudovirus), with viral entry detected by luciferase luminescence. The entry efficiency of pseudoviruses without any treatment was considered 100%. (**b**) Neutralization of authentic SARS-2 wild-type virus was assessed by incubating viral particles with antibody before adding to Vero HF cells. Viral infection was assessed by ELISPOT 24 hr after infection by immunostaining with the anti-SARS-2 nucleocapsid antibody 1C7C7. (**c**) ADCP was performed by co-incubating undifferentiated THP-1 cells, antibodies and pHrodo-Green/APC-polystyrene beads coated with HexaPro or 4P-DS. The phagocytosis score was calculated as the percent of positive APC/FITC cells multiplied by the GMFI for APC. Data were collected from two separate experiments with the average and standard deviation shown. (**d**) ADCC was assessed by incubating NK-92 V/V cells, HEK-293T cells transfected to express either wild-type SARS-2 spike (spike) or nothing (mock) and antibody. For each panel, data shown are representative of three biological replicates. Duplicate technical replicates with the midpoint of each condition are shown.

The online version of this article includes the following source data and figure supplement(s) for figure 5:

**Source data 1.** Data reporting antibody effect on infection with pseudovirus, authentic virus, phagocytosis score, and cellular lysis.

**Figure supplement 1.** Antibody 3A3 inhibits cellular fusion induced by the interaction of SARS-2 spike with human ACE2.

In addition to neutralization, antibodies binding spike can mediate Fc effector functions and thereby eliminate viral particles and infected cells. When SARS-2 HexaPro spike was coated on beads and incubated with S309, 3A3, or RAY53, the beads were effectively internalized by human THP-1 monocytes in an in vitro ADCP assay (*Figure 5c*). When 4P-DS was used, only S309 and RAY53 but not 3A3 or an isotype control mediated ADCP, consistent with RAY53's higher affinity to the exposed hinge epitope on 4P-DS. In an in vitro ADCC assay, HEK 293T cells transiently transfected with authentic SARS-2 spike were lysed by human NK-92 cells in the presence of control antibody CR3022 or RAY53, but not an isotype control (*Figure 5d*). These results indicate that RAY53 binding is compatible with ADCP and ADCC, although the impact of RAY53-mediated effector functions on disease progression requires future evaluation in animal models.

## Discussion

Antibodies binding S2 are an important component of the immune response to SARS-CoV-2 and other coronavirus infections. They are naturally elicited, with neutralizing sera from individuals never exposed to SARS-2 common in young people and exclusively binding the S2 domain (*Ng et al., 2020*). This S2 response is boosted upon exposure to SARS-2 (*Dowell et al., 2022*). These antibodies are generally

common in convalescent repertoires, with >300 anti-S2 sequences reported (*Raybould et al., 2021*), but with few detailed descriptions and just two epitope classes defined. Antibodies recognizing the SARS-2 S2 stem region between residues 1142–1160, proximal to the viral membrane, (*Hsieh et al., 2021*; *Pinto et al., 2021*; *Sauer et al., 2021*; *Ullah et al., 2021*; *Wang et al., 2021*; *Li et al., 2022*; *Zhou et al., 2022*) dominate linear S2 peptide responses (*Ladner et al., 2021*) but generally result in moderate-to-no neutralization of SARS-2 spike pseudovirus (10 to >300 nM IC$_{50}$). Antibodies binding the fusion peptide, which mediates steps required for viral entry of host cells, are highly cross-reactive to α- and β-coronavirus spike proteins, typically neutralize SARS-2 pseudoviruses (*Low et al., 2022*) and authentic viruses (*Dacon et al., 2022*) with modest in vitro IC$_{50}$ values >50 nM. Antibodies binding the hinge epitope in the S2 core are much less thoroughly described, although antibodies isolated from unexposed individuals that appear to bind near the hinge epitope based on negative staining EM images were enriched 37-fold upon infection (*Claireaux et al., 2022*). An emerging theme is that highly conserved S2 epitopes, whether located at the stem, fusion peptide or hinge, are immunogenic but often fail to mediate strong antibody neutralization (*Bowen et al., 2021*).

By contrast, antibodies targeting these highly conserved S2 epitopes seem to rely heavily on Fc effector functions to mediate protection by antibody-dependent viral phagocytosis, cellular cyto-toxicity, and/or trogocytosis of infected cells. The S2 stem binding antibodies S2P6 (*Pinto et al., 2021*) and IgG22 (*Hsieh et al., 2021*) protected SARS-2 challenged animals in vivo. IgG22 did not neutralize authentic virus in vitro, implicating Fc functions in protection in vivo, and S2P6 was shown to elicit ADCP and ADCC in vitro. Partial protection in a prophylaxis mouse model by stem-binding antibody CV3-25 was ablated by residues changes that silence the Fc (*Ullah et al., 2021*). Similarly, antibodies binding the fusion peptide moderately protected hamsters against severe disease (*Dacon et al., 2022*; *Low et al., 2022*). S2 core-binding antibodies require further validation, but those found in the human repertoire elicited ADCP and antibody-dependent cellular trogocytosis in vitro (*Claireaux et al., 2022*) while the hinge-binding antibody RAY53 reported here induced ADCP and ADCC in vitro (*Figure 5c and d*). Highly conserved S2 epitopes may be underappreciated targets based on simple neutralization assays, but valuable for eliciting effector functions against many vari-ants, similar to antibodies binding the flu stem region (*Impagliazzo et al., 2015*; *Corti et al., 2017*).

The presence of S2 core-binding antibodies in the naïve immune repertoire and their amplification after infection with pre-Omicron strains of SARS-2 suggests that S2 core epitopes exerted immune pressure in early waves of COVID-19 infection. Antibody RAY53's dramatic loss of binding to authentic SARS-2 Omicron BA.1 spike (*Figure 4d*) is consistent with evolutionary evasion of antibodies binding the S2 core as the virus has been repeatedly exposed to the human immune response. While anti-body evasion by the SARS-2 spike commonly occurs through substitutions within targeted epitopes (*Greaney et al., 2021*; *Starr et al., 2021*), SARS-2 Omicron BA.1 contains only one residue change within the hinge epitope which does not alter RAY53 binding in isolation (*Figure 3a*). Antibody RAY53 binds soluble Omicron BA.1 HexaPro spike, albeit with a depressed on-rate (*Figure 4c*, *Figure 4—figure supplement 4f*), but did not bind authentic Omicron BA.1 spike on the cell surface or neutralize the corresponding pseudovirus (*Figure 5a*). Omicron BA.1 appears to protect cryptic epitopes by closely packing the RBDs in the down state and possibly stabilizing the S2-closed state when RBDs are up for ACE2 binding. This structural protection may be evolutionarily favored over mutation when altering the epitope sequence carries a high functional cost, as for the hinge epitope (*Figure 3d*).

SARS-2 spike is a highly dynamic protein, sensitive to many variables including temperature (*Edwards et al., 2021*), pH (*Zhou et al., 2020*), and glycosylation (*Casalino et al., 2020*). Each of the RBDs has the potential to flip up, which has been captured in cryo-EM images of closed, one-up, two-up, and three-up spikes (*Benton et al., 2020*). Additionally, molecular dynamics simulations indicated that movement of the S1 domains during opening extends deeper into the protein than previously appreciated (*Zimmerman et al., 2021*), while HDX analysis has observed an open-S2 conformation with a splayed spike trimer that exposes core S2 epitopes (*Costello et al., 2022*). Notably, these movements impact hinge epitope accessibility, indicating that antibody binding to this region involves multiple kinetic steps (*Figure 2f*). Data showing enhanced binding to spike variants favoring the S2 open state (*Figure 2c–e*) and similar neutralization of authentic spike on pseudo-viruses by 3A3 and RAY53 (*Figure 5a*) despite the latter's increased affinity (*Figure 4e*) indicate that spike S2 opening is the rate-limiting step in antibody binding. Results shown here also underscore that these rates can vary depending on the local environment (soluble protein versus viral or infected cell surface) and

amino acid changes present (stabilizing changes versus circulating variants), highlighting the importance of spike dynamics on antibody binding.

Here, we report an S2 hinge epitope that is conserved across all highly pathogenic coronavirus strains and its interactions with two related antibodies. Immunization with 2P-stabilized spike and natural infection appears to elicit antibodies binding this epitope in the S2 core, but antibody access is restricted by spike dynamics that expose the epitope. Although targeting this epitope alone is unlikely to be potently neutralizing, strategies that enhance access to the highly conserved S2 core such as ACE2-mimicking antibodies (*Low et al., 2022*) may allow existing antibody repertoires to more effectively promote viral clearance by recruiting Fc effector functions. Future work will use these conformationally-selective antibodies to elucidate spike behavior in response to stabilizing and evolved mutations as well as environmental conditions including spike surface density, protease priming, and interactions with other cellular or viral membrane proteins.

# Materials and methods

## Key resources table

| Reagent type (species) or resource | Designation | Source or reference | Identifiers | Additional information |
|---|---|---|---|---|
| Antibody | Anti-NP 1C7C7 | Thomas Moran (The Icahn School of Medicine at Mount Sinai) | N/A | |
| Antibody | Anti-spike CR3022 | Constructed based on *ter Meulen et al., 2006*. | N/A | |
| Antibody | Anti-spike mAb 2–4 | Constructed based on *Liu et al., 2020* . | N/A | |
| Antibody | Anti-spike S309 | Constructed based on *Pinto et al., 2020*. | N/A | |
| Antibody | Anti-StrepTagII Fab (clone C23.21) | Constructed based on patent WO2015067768A1 (Institut Pasteur) | N/A | |
| Antibody | Antibody variants: 3A3, hu3A3, RAY53, 3E11 | This study | N/A | Sequences can be found in *Supplementary file 4*. |
| Antibody | Goat anti human κ HRP | SouthernBiotech | Cat# 2060-05 | |
| Antibody | Goat anti human IgG Fc-AF647 | Jackson ImmunoResearch | Cat# 109-605-008 | |
| Antibody | Goat anti mouse Ig HRP | SouthernBiotech | Cat# 1010-05 | |
| Antibody | Goat anti-human IgG Fc-HRP (polyclonal) | SouthernBiotech | Cat# 2047-05 | |
| Antibody | Human Fab$_2$ anti-strep-tag (clone C23.21) | Jason McLellan Lab | N/A | |
| Antibody | Mouse anti c-myc, clone 9E10 | BioXCell | Cat #MA1-980 | |
| Antibody | Mouse anti FLAG (M2) HRP | Sigma-Aldrich | Cat# A8592 | |
| Antibody | Mouse anti FLAG (M2) PE | BioLegend/Prozyme | Cat# 637309/ #PJ315 | |
| Antibody | Mouse anti-M13 pVIII-HRP, clone RL-ph1 | Santa Cruz Biotech | Cat# sc53004 | |
| Cell line (*Cricetulus griseus*) | CHO-T | Acyte BioTech | N/A | |
| Cell line (*C. griseus*) | CHOK-1 | ATCC | Cat# CCL-61 | |
| Cell line (*C. griseus*) | ExpiCHO | Thermo Fisher Scientific | Cat# A29133 | |
| Cell line (*Homo sapiens*) | Expi293 | Thermo Fisher Scientific | Cat# A41249 | |
| Cell line (*H. sapiens*) | Freestyle HEK293-F | Thermo Fisher Scientific | Cat# R79007 | |
| Cell line (*H. sapiens*) | HEK-293T-hACE2 | BEI Resources | Cat# NR-52511 | |

*Continued on next page*

*Continued*

| Reagent type (species) or resource | Designation | Source or reference | Identifiers | Additional information |
|---|---|---|---|---|
| Cell line (*H. sapiens*) | HEK293T | ATCC | Cat# CRL-3216 | |
| Cell line (*H. sapiens*) | NK-92 V158 | ATCC | Cat# PTA-8836 | |
| Cell line (*H. sapiens*) | THP-1 | ATCC | Cat# TIB-202 | |
| Cell line (*H. sapiens*) | Vero HL | *Piepenbrink et al., 2022* | N/A | |
| Chemical compound, drug | Biotin | Sigma-Aldrich | Cat# B4501-10G | |
| Chemical compound, drug | Calcein AM | BD Pharmingen | Cat# 564061 | |
| Chemical compound, drug | Flash Red 1μ Beads | Bangs Laboratories | Cat# FSFR004 | |
| Chemical compound, drug | pHrodo iFL Green STP Ester | Thermo Fisher Scientific | Cat# P36013 | |
| Chemical compound, drug | TMB Substrate | Thermo Fisher Scientific | Cat# 34021 | |
| Commercial assay or kit | Alexa Fluor 647 Protein Labelling Kit | Fisher Scientific | Cat# A20173 | |
| Commercial assay or kit | ExpiFectamine 293 Transfection Kit | Thermo Fisher Scientific | Cat# A14524 | |
| Commercial assay or kit | ExpiFectamine CHO Transfection Kit | Thermo Fisher Scientific | Cat# A29129 | |
| Commercial assay or kit | HiTrap Protein A columns | Cytiva | Cat# 17-5498-54P | |
| Commercial assay or kit | IMAC Sepharose 6 Fast Flow resin | Cytiva | Cat# 17092107 | |
| Commercial assay or kit | Lipofectamine 2000 | Thermo Fisher Scientific | Cat# 11668019 | |
| Commercial assay or kit | Mycostrip test | Invivogen | Cat# rep-mys-10 | |
| Commercial assay or kit | Octet Anti-Human Fab-CH1 2nd Generation (FAB2G) Biosensors | Forte Bio | Cat# 18-5125 | |
| Commercial assay or kit | Octet Streptavidin (SA) Biosensor | Forte Bio | Cat# 18-5019 | |
| Commercial assay or kit | Protein Thermal Shift Dye Kit | Thermo Fisher Scientific | Cat# 4461146 | |
| Commercial assay or kit | Series S Sensor Chip CM5 | Cytiva | Cat# BR100530 | |
| Commercial assay or kit | Strep-Tactin XT Superflow high capacity cartridge | IBA | Cat# 2-4026-001 | |
| Commercial assay or kit | Superdex 200 Increase 10/300GL | Cytiva | Cat# 28-9909-44 | |
| Organisms (*Mus musculus*) | Balb/c mice | Charles River | Cat# 028 | |
| Other | HBS EP+buffer | Cytiva | Cat# BR100669 | |
| Peptide, recombinant protein | Avidin | Sigma-Aldrich | Cat# A9275-25MG | |

*Continued on next page*

*Continued*

| Reagent type (species) or resource | Designation | Source or reference | Identifiers | Additional information |
|---|---|---|---|---|
| Peptide, recombinant protein | Streptavidin AF647 | Jackson ImmunoResearch | Cat# 016600084 | |
| Peptide, recombinant protein | Streptavidin PE | BioLegend | Cat# 405204 | |
| Recombinant DNA reagent | AbVec hIgG1 | *Smith et al., 2009* | N/A | |
| Recombinant DNA reagent | AbVec hIgKappa | *Smith et al., 2009* | N/A | |
| Recombinant DNA reagent | HDM-IDTSpike-fixK | BEI Resources | Cat# NR-52514 | |
| Recombinant DNA reagent | M13KO7 helper phage (virus) | NEB | N0315S | |
| Recombinant DNA reagent | pcDNA3.1(-)- Wuhan-Hu-1 Spike | *Walls et al., 2020* BEI Resources | Cat# NR-52420 | |
| Recombinant DNA reagent | pCMV-VSV-G | Cell Biolabs | Cat# RV-110 | |
| Recombinant DNA reagent | pCTCon-Fab | *Wang et al., 2018* | N/A | |
| Recombinant DNA reagent | pHAGE-CMV-Luc2-IRS-ZsGreen-W | BEI Resources | Cat# NR-52516 | |
| Recombinant DNA reagent | pHAGE2-EF1aInt-ACE2-WT | BEI Resources | Cat# NR-52512 | |
| Recombinant DNA reagent | pLEX307-DPP4-G418 | Addgene | Cat# 158453 | |
| Recombinant DNA reagent | pMoPac24 | *Hayhurst et al., 2003* | N/A | |
| Sequence-based reagent | Primers for cloning mouse variable regions | *Krebber et al., 1997* | N/A | |
| Software, algorithm | Astra Software V6.1.2 | Wyatt Technology | RRID:SCR_016255 | |
| Software, algorithm | Biacore X100 Evaluation Software V2.0.1 | GE Healthcare | N/A | |
| Software, algorithm | cisTEM | *Grant et al., 2018* | RRID:SCR_016502 | |
| Software, algorithm | cryoSPARC | *Punjani et al., 2017* | RRID:SCR_016501 | |
| Software, algorithm | DynamX v3.0 | Waters | Part# 720005145en | |
| Software, algorithm | Excel 1808 | Microsoft | N/A | |
| Software, algorithm | Fiji | *Schindelin et al., 2012* | RRID:SCR_002285 | |
| Software, algorithm | FlowJo 10.7.1 | BD Biosciences | RRID:SCR_008520 | |
| Software, algorithm | GraphPad Prism, v9.2.0 | *Motulsky and Brown, 2006* | RRID:SCR_002285 | |
| Software, algorithm | HD-eXplosion v 1.2 | Naifu Zhang and Sheena D'Arcy (The University of Texas at Dallas) | N/A | |
| Software, algorithm | Image J v1.53e | NIH | RRID:SCR_003070 | |
| Software, algorithm | Octet Data Analysis Software V11.1 | Forte Bio | N/A | |
| Software, algorithm | ViiA 7 Software | Thermo Fisher Scientific | N/A | |

*Continued on next page*

*Continued*

| Reagent type (species) or resource | Designation | Source or reference | Identifiers | Additional information |
|---|---|---|---|---|
| Strain, strain background (*Escherichia coli*) | DH5α electrocompetent cells | NEB | Cat# C2987H | |
| Strain, strain background (*E. coli*) | XL1-Blue | Agilent | Cat# 200228 | |
| Strain, strain background (*Saccharomyces cerevisiae*) | AYW101 | *Wentz and Shusta, 2007* | N/A | |
| Strain, strain background (*S. cerevisiae*) | EBY100 yeast | ATCC | Cat# MYA-4941 | |

## Nomenclature

In this work, 'spike' refers to the extracellular coronavirus fusogen domains containing homologous 2P changes (proline substitutions at residues 986 and 987 in SARS-CoV-2) C-terminally fused to a foldon domain (*Wrapp et al., 2020a*), whereas 'authentic' refers to spike variants as expressed without stabilizing changes as found on the virion, with other variations noted. 'Wild-type' SARS-CoV-2 spike refers to the spike sequence originally reported in January of 2020 for ancestral (Wuhan-Hu-1, GenBank accession number MN908947) SARS-CoV-2. For brevity and clarity, 'SARS-2' refers to the SARS-CoV-2 virus or spike, 'SARS-1' to SARS-CoV, and 'MERS' to MERS-CoV henceforth.

## Cell lines

Eukaryotic cell lines were obtained from the sources listed in the reagent table above under 'Cell lines.' No commonly misidentified cell lines were used in this study. Cell lines were purchased from reputable suppliers for protein expression and not further authenticated with the exception of HEK-293T-hACE2 cells, which were generated in the lab. These cells were verified to express human ACE2 after lentiviral infection and selection by Western blot as described in the text. Cell lines growing in the lab are tested approximately once per year for mycoplasma contamination using InvivoGen's MycoStrip test, with no contamination detected.

## Spike expression

Soluble coronavirus spikes and spike variants were expressed and purified as previously described (*Hsieh et al., 2020*; *Wrapp et al., 2020b*). SARS-2 (*Wrapp et al., 2020a*), SARS-1 (*Kirchdoerfer et al., 2018*), SARS-2 HexaPro (*Hsieh et al., 2020*), MERS (*Pallesen et al., 2017*), HKU1 (*Pallesen et al., 2017*), and variant spikes were expressed in ExpiCHO or Freestyle 293F cells (Thermo Fisher Scientific). MERS S2 included residues 763–1291 of MERS-2P with 8 additional stabilizing substitutions (*Hsieh et al., 2021*). SARS-2 HexaPro S2 included residues 697–1208 of the SARS-2 spike with an artificial signal peptide, proline substitutions at positions 817, 892, 899, 942, 986, and 987 and a C-terminal T4 fibritin domain, HRV3C cleavage site, 8×HisTag and TwinStrepTag. HexaPro RBD-locked-down (*Xiong et al., 2020*) included the substitutions S383C-D985C in SARS-2 HexaPro. Aglycosylated HexaPro was produced by treating SARS-2 HexaPro with Endo H overnight at 4°C leaving only one N-acetylglucosamine attached to N-glycosylation site.

## Murine immunization

Three 6-week-old female BALB/c (Charles River labs, Cat# 028) mice were immunized subcutaneously with 5 μg prefusion stabilized MERS S2 and 20 μg of ODN1826 + 100 μl of 2× Sigma Adjuvant System (SAS; Sigma) containing monophosphoryl lipid A and trehalose dimycolate in squalene oil. Four weeks later, the mice were boosted with the same dose of the same mixture. Three weeks after boosting, the mice were sacrificed and spleens were collected in RNALater (Thermo Fisher). All of the animals were handled according to approved institutional animal care and use committee (IACUC) protocols approved by the University of Texas at Austin (protocol AUP-2018-00092).

## Phage display antibody library construction

RNA from each mouse was isolated from the aqueous phase of homogenized spleens mixed with 1-bromo-3-chloropropane and purified with the PureLink RNA kit (Invitrogen) separately. The Superscript IV kit (Invitrogen) was used to synthesize cDNA. The $V_H$ and $V_L$ sequences from each immunized mouse were amplified with mouse-specific primers described by *Krebber et al., 1997*. Maintaining separate reactions for each mouse, the $V_L$ and $V_H$ regions were joined by overlap extension PCR to generate $V_L$-linker-$V_H$ fragments (scFv) in which the linker region encodes the amino acids $(Gly_4Ser)_4$ and *Sfi*I sites flanked the scFv sequence. The scFv PCR products were pooled and cloned into pMopac24 (*Hayhurst et al., 2003*) via *Sfi*I cut sites to encode an scFv with a c-terminal myc tag fused to the M13 phage pIII protein. This library was then transformed to XL1-Blue (Agilent Technologies) *Escherichia coli*. The total number of transformants was $3.1 \times 10^8$ with <0.01% background based on plating.

## Phage display and panning

The *E. coli* containing the library were expanded in growth media (2×YT with 1% glucose, 200 µg/mL ampicillin, 10 µg/mL tetracycline) at 37°C to an $OD_{600}$ of 0.5, then infected with $1 \times 10^{11}$ pfu/mL M13K07 helper phage (NEB) and induced with 1 mM isopropyl β-d-1-thiogalactopyranoside. After 2 hr of shaking at room temperature, 12.5 µg/mL of kanamycin was added for phage expression overnight. Phage were precipitated in 20% PEG-8000 in 2.5 M NaCl, titered by infection of XL1-Blue and plating, and used for Round 1 panning. This process was repeated for each round of panning, starting from overnight growth of the output phage from each round.

Four rounds of panning were used to isolate scFvs binding both MERS S2 and SARS-2 spike using the following solutions coated on high binding plates: 2 µg/mL anti-c-myc tag antibody (Invitrogen) to eliminate phage expressing no or truncated scFv (round 1), 2 µg/mL MERS S2 (round 2), 2 µg/mL SARS-2 spike (round 3), and 0.4 µg/mL SARS-2 spike (round 4). In each round of panning, the plates were blocked with 5% non-fat milk in phosphate-buffered saline (PBS) with 0.05% Tween-20 (PBS-T), and phage were preincubated with 5% non-fat milk in PBS-T for 30 min before incubation on the plate for 1.5 hr at room temperature. After thorough washing with PBS-T, output phage was eluted using 0.1 M HCl at pH 2.2, neutralized with ~1:20 2 M Tris base, and allowed to infect XL1-Blue cells overnight amplification.

Random clones isolated after rounds 3 and 4 of panning were sequenced and unique clones were tested by monoclonal phage enzyme-linked immunosorbent assay (ELISA) on plates coated with SARS-2 spike or RSV F foldon at 2 µg/mL in PBS. Briefly, plates were coated overnight at 4°C, washed with PBS-T, then blocked with PBS-T and 5% milk. Phage were allowed to bind for 1 hr at room temperature, thoroughly washed with PBS-T, then incubated with 1:2000 anti-M13 pVIII-HRP (GE Healthcare) in PBS-T 5% milk for another hour. After washing, the plate was developed with the TMB Substrate Kit (Thermo Scientific), quenched with an equal volume of 1 M HCl and evaluated by absorbance at 450 nm.

## Antibody expression, purification, and quality control

Full-length antibody versions of 3A3 and 3E11 were cloned as previously described (*Nguyen et al., 2015*) as mouse variable region-human IgG1 constant region chimeras. Antibodies hu3A3 and RAY53 were similarly cloned into human IgG1 and Ig$\mathrm{K}$ expression vectors. Antibodies were expressed in ExpiCHO (Thermo Fisher Scientific) cells according to the high titer protocol provided and purified on a Protein A HiTrap column (GE Healthcare) with the ACTA Pure FPLC system (GE Healthcare), and buffer exchanged to PBS.

Human Fab fragments were generated by digestion of full-length antibody with papain and removal of the Fc portion by protein A binding. Mouse Fab fragments of 3A3 were generated by cloning the $V_H$ and $V_L$ regions upstream of heavy chain constant regions with a HRV3C protease site in the hinge (*Pallesen et al., 2017*) and a mouse kappa chain, respectively. After expression, protein A purified protein was digested with HRV3C protease, and the flow-through from a protein A HiTrap column was collected. Excess HRV3C protease was removed by incubation with Ni Sepharose 6Fast Flow beads (GE Healthcare). Fully murine antibodies were produced by cloning the $V_H$ regions into mouse IgG2a and $V_L$ regions in to a mouse IgK expression cassettes in the pAbVec background, co-transfecting, and purifying as described above.

## Hydrogen-deuterium exchange mass spectrometry

Hydrogen-deuterium exchange was performed on complexes were formed with excess antibody (0.50 µM SARS-2 HexaPro spike protein alone or in the presence of 0.55 µM 3A3 IgG or Fab) such that the SARS-2 HexaPro spike was expected to be ~90% bound based on the known protein concentrations and measured $K_d$. Only spike protein peptides were analyzed in this experiment. Complexes were thawed from –80°C storage on ice and incubated for 10 min at 25°C before exchange in 90% deuterium and 20 mM Tris pH 8.0, 200 mM NaCl. The exchange was quenched after $10^1$, $10^2$, $10^3$, and $10^4$ s by mixing samples 1:1 with cooled 0.2% (v/v) formic acid, 200 mM TCEP, 8 M Urea, pH 2.3. Samples were immediately flash-frozen in liquid $N_2$ and stored at –80°C. Hydrogen-deuterium exchange was similarly performed on 0.50 µM 3A3 IgG alone or in the presence of 0.75 µM of SARS-2 HexaPro spike protein. Only 3A3 IgG peptides were analyzed in this experiment. Samples were prepared as described above, but in 86% deuterium for $10^1$, $10^2$, and $10^3$ s.

Samples were thawed and LC-MS performed using a Waters HDX manager and SYNAPT G2-*Si* Q-Tof. Three or four technical replicates of each sample were analyzed in random order. Samples were digested online by *Sus scrofa* Pepsin A (Waters Enzymate BEH Pepsin column) at 15°C and peptides trapped on a C18 pre-column (Waters ACQUITY UPLC BEH C18 VanGuard pre-column) at 1°C for 3 min at 100 µL/min. Peptides were separated over a C18 column (Waters ACQUITY UPLC BEH C18 column) and eluted with a linear 3–40% (v/v) Acetonitrile gradient for 7 min at 30 uL/min at 1°C and 0.1% (v/v) formic acid as the basic LC buffer.

MS data were acquired using positive ion mode and either HDMS or HDMS$^E$. HDMS$^E$ mode was used to collect both low (6 V) and high (ramping 22–44 V) energy fragmentation data for peptide identification in water-only samples. HDMS mode was used to collect low-energy ion data for all deuterated samples. All samples were acquired in resolution mode. The capillary voltage was set to 2.8 kV for the sample sprayer. Desolvation gas was set to 650 L/hr at 175°C. The source temperature was set to 80°C. Cone and nebulizer gas were flowed at 90 L/hr and 6.5 bar, respectively. The sampling cone and source offset were set to 30 V. Data were acquired at a scan time of 0.4 s with a range of 100–2000 m/z. A mass correction was done using [Glu1]-fibrinopeptide B as a reference mass.

Water-only control samples were processed by Protein Lynx Global Server v.3.0.2 with a 'minimum fragment ion matches per peptide' of 3 and allowing methionine oxidation. The low and elevated energy thresholds were 250 and 50 counts, respectively, and the overall intensity threshold was 750 counts. The resulting peptide lists were then used to search data from deuterated samples using DynamX v.3.0. We did not search for glycosylated peptides as de-glycosylation had little impact on 3A3 binding (*Figure 4—figure supplement 4f*). Peptide filters of 0.3 products per amino acid and one consecutive product were used. Spectra were manually assessed, and figures were prepared using HD-eXplosion (*Zhang et al., 2020*) and PyMOL (*DeLano, 2002*). The HDX data summary table (*Supplementary file 1*) and complete data table (*Supplementary file 2*) are included. The location of the 3A3 epitope was confirmed in a separate experiment carried out over the temperature range of 4–37°C (*Costello et al., 2022*).

## Low-resolution cryo-EM

To form spike-antibody complex, prefusion-stabilized SARS-CoV-2 S2 was incubated with 1.5-fold molar excess of 3A3 Fab at room temperature for 20 min. The mixture was then applied to a size-exclusion column (SEC) in a running buffer containing 2 mM Tris pH 8.0, 200 mM NaCl, and 0.02% NaN$_3$ to obtain a peak fraction containing the S2-3A3 Fab complex for cryo-EM sample preparation. The complex at 0.5 mg/mL was deposited on a plasma-cleaned Au-Flat 1.2/1.3 grid (ProtoChip), which was plunge-frozen using a Vitrobot Mark IV (Thermo Fisher) with 4 s blot time and –2 force at 100% humidity at 22°C. A total of 1179 micrographs were collected using a Glacios (Thermo Fisher) equipped with a Falcon IV direct electron detector. Data were collected at a magnification of 150,000×, corresponding to a calibrated pixel size of 0.94 Å/pix. CryoSPARC v3.2.0 was used for patch motion correction, CTF estimation, particle picking, and particle curation via iterative rounds of 2D classification (*Punjani et al., 2017*). One class that had the best-resolved 3D reconstruction from heterogenous refinement was used for subsequent non-uniform homogeneous refinement. ChimeraX (*Pettersen et al., 2021*) was used to generate a mask that encompassed the Fab and the apex of an S2 protomer to perform focused refinement. A protomer of SARS-CoV-2 spike (PDB: 6XKL) without the S1 subunit and an ABodyBuilder-predicted 3A3 Fab structure (*Dunbar et al.,*

*2016*) was used as a model to dock into the local EM map generated by focused refinement using cryoSPARC v3.2.0.

## Biolayer interferometry (BLI) and surface plasmon resonance (SPR) measurements

To evaluate ACE2 binding to HexaPro captured by 3A3, AHC anti-human IgG Fc (ForteBio) sensors were used to pick up 3A3 (10 nM) to a response of 0.6 nm. Then mAb-coated tips were dipped into wells containing HexaPro (50 nM) to a response of 0.6 nm and then dipped into wells containing ACE2 (50 nM), irrelevant murine mAb (50 nM), or buffer. Association of mu3A3/irrelevant mAb was measured for 5 min and dissociation for 10 min.

To compare 3A3 and mAb 2–4 binding to HexaPro and 'Down' HexaPro, AHC anti-human IgG Fc (ForteBio) sensors were loaded with 3A3 or mAb 2–4 mAb in the kinetics buffer at 10 nM to a response of 0.6 nm. After a baseline step, the sensors were incubated with either HexaPro or 'Down' HexaPro, both at 60 nM for 5 min. Dissociation step was recorded for 10 min in the kinetics buffer.

To determine the affinity of 3A3 Fab by BLI, AHC anti-human IgG Fc (ForteBio) sensors were coated with the anti-foldon antibody identified in this work (3E11) at 10 nM in the kinetics buffer (0.01% BSA and 0.002% Tween-20 in PBS) to a response of 0.6 nm. MAb-coated sensors were then incubated with HexaPro S2 at 60 nM to a response of 0.6 nm. Association of 3A3 Fab was recorded for 5 min in kinetics buffer, starting at 100 nM followed by 1:2 dilutions. The dissociation was recorded for 10 min in the kinetics buffer. $K_d$ values were obtained using a 1:1 global fit model using the Octet instrument software. 3A3 Fab kinetics measurement was repeated once.

To determine the on-rate ($k_{on}$) values for 3A3 and RAY53 binding to various spike constructs, AHC anti-human IgG Fc (ForteBio) sensors were loaded with 10 nM antibody in kinetics buffer to a response of 0.6 nm. Association curves were recorded by incubating the sensors in spike, serial diluted 1:2. The dissociation step was recorded in the kinetics buffer without spike. On-rate values were determined using the 1:1 association non-linear fit on GraphPad prism 9.4.1 with off-rates constrained to $1 \times 10^{-12}$ $s^{-1}$.

For all BLI experiments, an Octet Red96 (ForteBio) instrument was used. Between every loading step, sensors were washed with kinetics buffer for 30 s. Before use, sensors were hydrated in the kinetics buffer for 10 min. After each assay, the sensors were regenerated using 10 mM Glycine, pH 1.5.

SPR was used to determine the binding kinetics and equilibrium affinity of the 3A3 Fab and HexaPro S2 interaction as well as the RAY53 and 4P-DS. An anti-StrepTagII Fab (clone C23.21) was covalently coupled to a CM5 sensor chip in 10 mM sodium acetate at pH 4.0 for a final RU of ~4300. It was then used to capture purified SARS-2 HexaPro S2 or 4P-DS by the c-terminal twin StrepTag to ~80 or ~1000 response units (RU), respectively, in each cycle using a Biacore X100 (GE Healthcare). The binding surface was regenerated between cycles using 0.1% SDS followed by 10 mM glycine at pH 2. The IgG or Fab was serially diluted and injected over the blank reference flow cell and then SARS-2 HexaPro S2- or 4P-DS-coated flow cell in HBS-P+ buffer. Buffer was also injected through both flow cells as a reference. The data were double-reference subtracted and fit to a 1:1 binding model using BIAevaluation software.

## ELISA evaluation of antibody binding

ELISAs were in either a spike or antibody capture configuration as indicated. For spike capture, plates were coated with 1 µg/mL of purified spike proteins in PBS. Duplicate serial dilutions of each full-length antibody in PBS-T with 5% milk were allowed to bind each coat, and the secondary antibody solution was a 1:1200 dilution of goat-anti-human IgG Fc-HRP (SouthernBiotech). For antibody capture, antibody was coated at 1 µg/mL in PBS. Duplicate serial dilutions of spike in PBS-T with 3% bovine serum albumin were allowed to bind each coat, and the secondary antibody solution was a 1:2000 dilution of streptactin-HRP (IBA Lifesciences). ELISA curves were fit to a four-parameter logistic curve.

Fresh aliquots of SARS-2 and SARS-2 HexaPro spikes were thawed and split to stress the spike proteins. One half of the aliquot was stressed by incubation at –20°C for 5 min, then 50°C for 2 min for three temperature cycles. The freshly thawed and stressed spikes were evaluated in antibody capture ELISAs. For each fresh and stressed spike, 8 µg was analyzed by SDS-PAGE under non-reducing conditions.

## Humanization of 3A3

Humanized 3A3 $V_H$ and $V_L$ regions were designed as previously described (*Nguyen et al., 2015*), and the variable regions were cloned into human IgK and IgG1 expression plasmids. The heavy and light chains were transfected together into ExpiCHO cells for combinatorial analysis of expression level and HexaPro binding. The veneering method of humanization for both $V_L$ and $V_H$ resulted in binding equivalent to 3A3 with slightly improved expression.

## Yeast display and engineering of 3A3

The hu3A3 light chain and Fab heavy chain region were cloned into pCTCON-Fab (*Wang et al., 2018*) with the heavy chain fused to Aga2 and a c-Myc tag and c-terminal FLAG tag on the light chain. The $V_H$ and $V_L$ regions were subjected to random mutagenesis at a target rate of 0.3%. In parallel, a site-directed library was created using primers encoding degenerate codons at CDRL2 and CDRH3 locations implicated in spike binding by HDX. Both the random and site-directed PCR products were integrated into *Saccharomyces cerevisiae* strain AWY101 (*Wentz and Shusta, 2007*) yeast plasmid by homologous recombination as previously described (*Benatuil et al., 2010*), resulting in approximately $3 \times 10^7$ variants in each library. Libraries and transformed yeast were grown and maintained in YNB media with casamino acids and 2% glucose at 30°C.

To induce expression of surface displayed Fab, yeast were subcultured to an $OD_{600}$ of 0.5 in YNB media with casamino acids, 0.2% glucose and 1.8% galactose and allowed to grow for 24 hr at 25°C. Libraries were sorted for three or four rounds by staining with 1:200 anti-FLAG-R-PE (ProZyme) and 50 nM 4P-DS directly labeled with Alexa Fluor 647 (Thermo Fisher) for 15 min at room temperature and 45 min on ice, and the brightest AF647 cells also fluorescent in the PE channel were sorted on a SONY MA900 cell sorter. Individual clones were isolated and the variable regions from the most promising 4P-DS binding yeast clones were amplified and transferred to mammalian expression vectors. Heavy and light chain candidates were transfected for combinatorial screening and evaluated for binding to 4P-DS by ELISA. RAY53 was the highest binding clone isolated and is comprised of a light chain variable region originating from the site-directed library and a heavy chain from the random mutagenesis library.

## Western blot of antibody binding to coronavirus spike proteins

Purified coronavirus spike proteins were reduced and boiled, and 50 ng of each was subjected to SDS-PAGE and transfer to PVDF membranes in duplicate. After blocking with PBS-T with 5% milk, the membranes were probed with 0.2 µg/mL 3A3 or 3E11 for 1 hr at room temperature. After washing with PBS-T, the membranes were incubated with 1:4000 goat anti-human IgG Fc-HRP for 45 min at room temperature, then developed with the SuperSignal West Pico Chemiluminescent Substrate (Thermo Scientific) and imaged.

## Mammalian expression and lentiviral plasmids

Plasmids required for mammalian expression and lentiviral production were obtained from BEI Resources. Plasmids expressing the HIV virion under the CMV promotor (HDM-Hgpm2, pRC-CMV-Rev1b, and HDM-tat1b) were provided under the following catalog numbers NR-52517, NR-52519, and NR-52518, respectively (*Crawford et al., 2020*). Plasmids for lentiviral backbone expressing a luciferase reporter under the CMV promotor followed by an IRES and ZsGreen (pHAGE-CMV-Luc2-IRS-ZsGreen-W) or human ACE2 gene (GenBank ID NM_021804) under an EF1a promoter (pHAGE2-EF1aInt-ACE2-WT) were provided as NR-52516 and NR52512, respectively (*Crawford et al., 2020*). The envelop vector expressing a codon-optimized wild-type SARS-2 spike protein (GenBank ID NC_045512) under a CMV promoter was obtained from BEI resources (HDM-IDTSpike-fixK, NR-52514, called pWT-SARS-2-spike here) (*Crawford et al., 2020*) The lentiviral backbone vector expressing a human DPP4 gene under an EF1a promoter (pLEX307-DPP4-G418) was obtained from Addgene, while the plasmid expressing VSV G (vesicular stomatitis virus glycoprotein) was purchased from Cell Biolabs (pCMV-VSV-G, Part No. RV-110). The pWT-SARS-2-spike plasmid was employed as a template for site-directed mutagenesis to generate the expression plasmid for the D614G and D614G with D985L, E988Q, or E988A. SARS-2 Omicron BA.1 (B.1.1.529) Spike Gene ORF cDNA was purchased from SinoBiological Inc SARS-2 Omicron BA.1, SARS-1 and MERS spike sequences were cloned into the pWT-SARS-2-spike plasmid for pseudovirus production.

## Flow cytometric evaluation of antibody binding to mammalian surface displayed spike

On day 0, Expi-293 cells (Thermo Fisher) were transfected with pEGFP alone or pEGFP and pWT-SARS-2-spike in a 1:1 ratio. On day 2, RAY53 in concentrations ranging from 300 nM to 3.5 nM was used to stain $\sim 3 \times 10^5$ transfected cells for 1 hr on ice. All cells were collected, washed with PBS with 1% FBS, then incubated with 1:250 goat-anti-human Fc-PE for 1 hr on ice. Cells were washed again, then scanned for EGFP and PE fluorescence on a BD Fortessa flow cytometer, and analyzed with FlowJo. Cells were gated by FSC and SSC, singlets, then EGFP expression to only analyze transfected cells. The PE GMFI of the EGFP expressing cells at each concentration was then used to calculate the effective $K_d$ as described (*Feldhaus and Siegel, 2004*).

To assess RAY53 binding to additional spike variants, Expi-293 cells were transfected with either pWT-SARS-2-spike, pOmicronBA1-SARS-2-spike, pSARS-spike, or pMERS-spike (no pEGFP plasmid was used) and treated as described above. Either 100 nM of RAY53 or 10 nM S309 was used to stain the transfected cells before incubation with goat-anti-human Fc-PE.

## Confocal cell fusion assay

On day 0, the CHO-T cells (Acyte Biotech) were transfected with either pPyEGFP (*Nguyen et al., 2018*) or 1:4 pWT-SARS- CoV-2-spike:pPyEGFP using Lipofectamine 2000 (Life Technologies), and media was replaced on day 1. On day 2 after transfection, HEK-293T-hACE2 cells (BEI, NR-52511), which stably expresses human ACE2, were stained with 1 µM CellTrace Far Red dye (Invitrogen, Ex/Em: 630/661 nm) in PBS for 20 min at room temperature, then quenched with DMEM with 10% heat-inactivated FBS for 5 min, and resuspended in fresh media. CHO-T cells expressing EGFP or EGFP and surface spike were preincubated with the antibody for 1 hr at 37°C, then mixed with HEK-hACE2 cells at a ratio of 5:1 in 24-well plates with a coverslip on the bottom of each well. On day 3, after 20 hr of coincubation, the coverslip with bound cells was washed once with PBS and fixed with 4% paraformaldehyde for 20 min at room temperature, washed again, and mounted on slides with DAPI-fluoromount-G (SouthernBiotech). Images were collected with Zeiss LSM 710 confocal microscope (Carl Zeiss, Inc) and processed using ImageJ software (http://rsbweb.nih.gov/ij).

Two different statistical analysis methods determined the cell fusion level. The first statistical analysis was based on the percentage of HEK-ACE2 pixels (red) colocalizing with spike expressing CHO pixels (green), which was determined by the following equation within the JACoP plugin for ImageJ (*Bolte and Cordelières, 2006*):

$$\text{HEK} - \text{ACE2 colocalization\%} = \frac{(\textit{summed intensities at 633 nm wavelength of HEK}-\textit{ACE2 pixels colocalizaing with CHO pixels})}{(\textit{summed intensities at 633 nm wavelength of HEK}-\textit{ACE2 pixels})}$$

The colocalization percentage for each independent image was determined using the Manders' coefficient. The average HEK-ACE2 cell size after coincubation with CHO cells was also determined using ImageJ software. The images collected at 633 nm emission (red fluorescence) were converted into 16-bit grayscale and the threshold adjusted to highlight the cell structure. The average cell size was automatically counted using 'Analyze Particles' with a size threshold (50–infinity) to exclude the background noise. The cells on the edge were excluded. The statistical significance of either HEK-ACE2 colocalization percentage or average cell size between different conditions was calculated with ANOVA using GraphPad Prism 7 (GraphPad Software). Values represent the mean and standard deviation of at least 160 cells.

## Generation of HEK293T-ACE2 and HEK293T-DPP4 target cells

A lentiviral vector expressing human ACE2 (pHAGE2-EF1aInt-ACE2-WT) or DPP4 (pLEX307-DPP4-G418) an EF1a promoter was used to transduce HEK293T cells. Clonal selection depended on the susceptibility to infection by the pseudotyped lentiviral particles; selected clones were validated using Western blotting.

## SARS-2 spike-mediated pseudovirus entry assay

HIV particles pseudotyped with wild-type or the Omicron BA.1 variant of SARS-2 spike, SARS-1 spike, MERS spike, and VSV-G were generated in HEK 293T cells. A detailed protocol for generating these

particles was reported by *Crawford et al., 2020*. HEK 293T cells were co-transfected with plasmids for (1) HIV virion-formation proteins (HDM-Hgpm2, pRC-CMV-Rev1b, and HDM-tat1b); (2) lentiviral backbone expressing luciferase reporter (pHAGE-CMV-Luc2-IRES-ZsGreen-W); and (3) a plasmid encoding one of the envelope proteins (wild-type SARS-2, SARS-2 Omicron BA.1, SARS-1, MERS, or VSV G). 72 hours post-transfection, media containing the pseudovirus particles were collected, filtered, fractionated, and stored at –80°C. In all the assays, 10,000 target cells were seeded in each well of the 96-well plate and allowed to adhere overnight before virus treatment. For the SARS-2-spike mutagenesis studies, virus titer was estimated for each virus using the qPCR Lentivirus Titer Kit (abm LV900), following the manufacturer's protocol. An equal number of viral particles carrying each spike mutant were serially diluted and added directly to HEK293T-ACE2 target cells (in triplicate). For the neutralization assays, the particles were used directly in cell entry experiments or after pre-incubation with each antibody for one hour at room temperature or at 4°C for the viral particles pseudotyped with the MERS spike. After 60–72 hr, a total number of cells per well were estimated using IncuCyte ZOOM equipment with a ×10 objective. Then cells were treated with the Bright-Glo Luciferase Assay reagent (Promega, E2610) to detect a luciferase signal (relative luciferase units or RLU) following the manufacturer's protocol. The percentage of entry was estimated as the ratio of the relative luciferase units recorded in the presence and absence of the tested antibody and a half-maximal inhibitory concentration ($IC_{50}$) calculated using a three-parameter logistic regression equation (GraphPad Prism v9.0).

## Live SARS-2 viral neutralization assays

Approximately 200 PFU/well of SARS-2 WA-1 strain containing twofold dilutions (starting concentration 670 nM) of antibody were incubated at 37°C for 1 hr. Vero HL cells ($4 \times 10^4$ cells/well in quadruplicate) were infected with the virus/antibody mixture, or virus alone mixture for 1 hr. After 1 hr virus adsorption, the media was changed with post-infection media containing 2% FBS, 1% Avicel and antibody. At 24 hr post-infection, infected and mock infected control cells were fixed with 10% neutral formalin for 24 hr and were immunostained with the anti-NP monoclonal 1C7C7 antibody. Virus neutralization was evaluated and quantified using ELISPOT, and the percentage of infectivity calculated using sigmoidal dose–response curves. In both cases, mock-infected cells and viruses in the absence of antibody were used as internal controls. Dotted line indicates 50% neutralization. Data were expressed as mean and SD.

## Fc-dependent ADCP and ADCC assays

To assess the ability of antibodies to induce phagocytosis, Flash Red polystyrene beads (Bangs Laboratories) were coated with SARS-2 HexaPro or 4P-DS spike and stained with pHrodo Green (Thermo Fisher Scientific). The beads were incubated with 50,000 undifferentiated THP-1 cells at a ratio of 50:1, and antibodies at 3.4 nM for 4 hr at 37°C. After washing, 10,000 cells per sample were evaluated by flow cytometry on a BD Fortessa instrument for red (APC channel) and pHrodo Green fluorescence. The phagocytosis score was calculated as the percent of total cells fluorescent in both the APC and pHrodo Green channel multiplied by the GMFI for the APC channel (*Ackerman et al., 2011*). Data was collected from two separate experiments with two technical replicates each.

To evaluate ADCC, HEK-293T cells were transfected with pWT-SARS-2-spike or nothing (mock) and allowed to express surface spike for 2 days. The HEK-293T cells were then stained with 2 μM calcein AM (BD Biosciences) for 30 min in serum-free media at 37°C, washed thoroughly and incubated for 4 hr with 67 nM antibody and NK-92 V158 cells (ATCC) at a ratio of 10:1. The cells were then spun down. The fluorescence of the supernatant was measured with 488 nm excitation and 515 nm emission. Controls included HEK-293T cells alone (spontaneous release) and fully lysed with detergent (maximum lysis). The following calculation determined the percent lysis for each antibody:

$$\%lysis = \frac{sample\ RFU - spontaneous\ release\ RFU}{maximum\ lysis\ RFU - spontaneous\ release\ RFU}.$$

Three assays were performed in total with duplicate technical replicates.

## Statistical analyses

The means ± SD were determined for all appropriate data. For the mammalian cell fusion experiments, pseudovirus neutralization experiments and epitope variant analysis, a one-way analysis of

variance (ANOVA) with Tukey's simultaneous test with p-values was used to determine statistical significance between groups. Welch's *t*-test was used to determine the significance of deuterium uptake differences.

## Acknowledgements

The following reagent was contributed by David Veesler for distribution through BEI Resources, NIAID, NIH: Vector pcDNA3.1(-) containing the SARS-Related Coronavirus 2, Wuhan-Hu-1 Spike Glycoprotein Gene, NR-52420 (*Walls et al., 2020*). The following reagents were obtained through BEI Resources, NIAID, NIH: Human Embryonic Kidney Cells (HEK-293T) Expressing Human Angiotensin-Converting Enzyme 2, HEK-293T-hACE2 Cell Line, NR-52511; SARS-Related Coronavirus 2, Wuhan-Hu-1 Spike-Pseudotyped Lentiviral Kit, NR-52948 (including NR-52514, NR-52516, NR-52517, NR-51518, and NR-52519); and Vector pHAGE2 Containing the ZsGreen Gene, NR-52520. The authors would like to thank Erin Scherer for the plasmids encoding SARS-2 Delta HexaPro stabilized spike; Kamyab Javanmardi, Thomas H Segall-Shapiro, and Jimmy D Gollihar for the plasmid encoding SARS-2 Omicron BA.1 HexaPro stabilized spike; Thomas Moran for antibody 1C7C7; Alejandro Chavez and Sho Iketani for the plasmid pLEX307-DPP4-G418 (Addgene plasmid #158453); and Eric Shusta for the yeast strain AWY101. The authors would like to thank Gregory C Ippolito, Jason J Lavinder, Ilya Finkelstein, and George Delidakis for useful discussions and advice related to this work. Flow cytometry and confocal microscopy was performed at the Center for Biomedical Research Support Microscopy and Imaging Facility at UT Austin (RRID# SCR_021756). The authors gratefully acknowledge all data contributors for generating the SARS-CoV-2 genetic sequence and metadata and sharing them via the GISAID Initiative, on which some of our analysis was based. This work was supported by NIH grants AI127521 (JSM), GM133751 (SD), and AI122753 (JAM), the Bill & Melinda Gates Foundation INV-017592 (JSM and JAM); Welch Foundation grants F-1767 (JAM), F-0003-19620604 (JSM), F-1390 (KD), and AT-2059-20210327 (SD); NSF RAPID 2027066 (JAM), NSF GRFP (ANQ and AMD); and a University of Texas at Austin Texas Biologics grant to JAM and JSM. This research was supported, in part, by the UT System Proteomics Network (SD). SM is a Chan Zuckerberg Biohub Investigator.

## Additional information

### Competing interests

Rui P Silva, Yimin Huang, Annalee W Nguyen: is an inventor on U.S. patent application no. 63/135,913 ("Cross-reactive antibodies recognizing the coronavirus spike S2 domain"). Ching-Lin Hsieh: is an inventor on U.S. patent application no. 63/135,913 ("Cross-reactive antibodies recognizing the coronavirus spike S2 domain"). inventors on patent no. WO/2021/243122 and PCT/US2021/034713 ("Engineered Coronavirus Spike (S) Protein and Methods of Use Thereof"). Is an inventor on U.S. patent application no. 63/188,813 ("Stabilized S2 Beta-coronavirus Antigens'). Andrea M DiVenere: inventor on patent no. WO/2021/243122 and PCT/US2021/034713 ("Engineered Coronavirus Spike (S) Protein and Methods of Use Thereof"). Jason S McLellan: inventor on U.S. patent application no. 63/135,913 ("Cross-reactive antibodies recognizing the coronavirus spike S2 domain"). J.S.M. is an inventor on U.S. patent application no. 62/412,703 ("Prefusion Coronavirus Spike Proteins and Their Use") and U.S. patent application no. 62/972,886 ("2019-nCoV Vaccine"). Is an inventor on patent no. WO/2021/243122 and PCT/US2021/034713 ("Engineered Coronavirus Spike (S) Protein and Methods of Use Thereof"). Is an inventor on U.S. patent application no. 63/188,813 ("Stabilized S2 Beta-coronavirus Antigens'). Jennifer A Maynard: inventor on U.S. patent application no. 63/135,913 ("Cross-reactive antibodies recognizing the coronavirus spike S2 domain") and WO/2021/243122 and PCT/US2021/034713 ("Engineered Coronavirus Spike (S) Protein and Methods of Use Thereof"). The other authors declare that no competing interests exist.

## Funding

| Funder | Grant reference number | Author |
| --- | --- | --- |
| National Institutes of Health | AI127521 | Jason S McLellan |
| National Institutes of Health | GM133751 | Susan Marqusee |
| National Institutes of Health | AI122753 | Jennifer A Maynard |
| Bill and Melinda Gates Foundation | INV-017592 | Jason S McLellan Jennifer A Maynard |
| Welch Foundation | F-1767 | Jennifer A Maynard |
| Welch Foundation | F-0003-19620604 | Jason S McLellan |
| Welch Foundation | F-1390 | Kevin N Dalby |
| Welch Foundation | AT-2059-20210327 | Sheena D'Arcy |
| National Science Foundation | 2027066 | Jennifer A Maynard |
| National Science Foundation | graduate fellowship | Andrea M DiVenere |
| Chan Zuckerberg Initiative | investigator | Susan Marqusee |
| UT Austin Texas Biologics | grant | Jason S McLellan Jennifer A Maynard |
| UT System Proteomics Network | support | Sheena D'Arcy |

The funders had no role in study design, data collection and interpretation, or the decision to submit the work for publication.

## Author contributions

Rui P Silva, Formal analysis, Investigation, Methodology, Writing – review and editing; Yimin Huang, Ahlam N Qerqez, Investigation, Methodology, Writing – review and editing; Annalee W Nguyen, Conceptualization, Supervision, Validation, Investigation, Visualization, Methodology, Writing – original draft, Writing – review and editing; Ching-Lin Hsieh, Conceptualization, Investigation, Methodology, Writing – review and editing; Oladimeji S Olaluwoye, Jun-Gyu Park, Ahmed M Khalil, Amanda L Bohanon, Sophie R Shoemaker, Shawn M Costello, Eduardo A Padlan, Investigation; Tamer S Kaoud, Rebecca E Wilen, Investigation, Methodology; Laura R Azouz, Methodology; Kevin C Le, Andrea M DiVenere, Yutong Liu, Alison G Lee, Dzifa A Amengor, Resources; Susan Marqusee, Kevin N Dalby, Supervision, Writing – review and editing; Luis Martinez-Sobrido, Supervision, Investigation, Writing – review and editing; Sheena D'Arcy, Supervision, Visualization, Writing – review and editing; Jason S McLellan, Supervision, Funding acquisition, Writing – review and editing; Jennifer A Maynard, Conceptualization, Supervision, Funding acquisition, Visualization, Writing – original draft, Project administration, Writing – review and editing

## Author ORCIDs

Yimin Huang  http://orcid.org/0000-0003-4348-5688
Annalee W Nguyen  http://orcid.org/0000-0003-1268-7164
Oladimeji S Olaluwoye  http://orcid.org/0000-0002-7888-0672
Tamer S Kaoud  http://orcid.org/0000-0003-1298-8725
Kevin C Le  http://orcid.org/0000-0001-5098-7737
Amanda L Bohanon  http://orcid.org/0000-0002-3371-3416
Susan Marqusee  http://orcid.org/0000-0001-7648-2163
Kevin N Dalby  http://orcid.org/0000-0001-9272-5129
Sheena D'Arcy  http://orcid.org/0000-0001-5055-988X
Jennifer A Maynard  http://orcid.org/0000-0002-0363-8486

### Ethics

All of the animals were handled according to approved institutional animal care and use committee (IACUC) protocols approved by The University of Texas at Austin (protocol AUP-2018-00092).

### Decision letter and Author response

Decision letter https://doi.org/10.7554/eLife.83710.sa1
Author response https://doi.org/10.7554/eLife.83710.sa2

---

## Additional files

### Supplementary files

• Supplementary file 1. HDX summary table for antibody peptides.

• Supplementary file 2. Complete HDX data for spike and antibody peptides.

• Supplementary file 3. Width of isotopic distributions for example spike peptides. Peptides were selected from regions identified as having bimodal distributions in Costello et al. We had no coverage in residues 626–636 and 1146–1166. Peak width (PW) in Da was calculated in triplicate for each peptide in the non-deuterated sample (ND Control) and at each time point of exchange. The SD is reported. The change in peak width (ΔPW) was calculated by subtracting the PW of the control from the PW of the sample. Bimodality was assessed by taking the maximum peak width for a particular peptide (ΔPWmax) and assessing if it was greater than 2 Da. Peak width was calculated using a method similar to *Weis et al., 2006*. Peptides were centroided with the Apex3D algorithm using DynamX (Waters). Following manual curation, ion stick data were transferred into Excel as two columns of data, m/z values and intensities, and the maximum peak in the isotopic envelope was determined. The list was then searched in descending m/z order to identify the two lowest m/z peaks that straddled 20% of the maximum peak intensity. The m/z value at an envelope intensity of 20% of the maximum intensity was determined using linear interpolation between these two peaks. This process was repeated with a search in ascending m/z order. The relative peak width was determined by multiplying by z (the charge state). For peptide spectra without peaks straddling 20% of the maximum peak intensity on one side of the maximum, typical for lower m/z peaks for peptides exhibiting low deuteration, the farthest isotopic centroid peak on that side was used as the m/z limit for calculating peak width while the other m/z limit was determined using the previously described method.

• Supplementary file 4. Antibody variable region sequences.

• MDAR checklist

### Data availability

The authors declare that all data supporting the findings of this study are available within the article and its supplementary information files. Sequences of the novel antibodies reported (3A3, RAY53 and 3E11) are provided in *Supplementary file 4*.

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
