## [Editor Report]

This study presents valuable findings on the isolation of an antibody that recognizes a novel, highly conserved SARS-CoV-2 Spike (S) epitope, adding to the growing repertoire of anti-S antibodies cross-reactive against human and zoonotic coronaviruses. The authors provide solid evidence supporting their claims, although the proposed antiviral mechanism of the newly described antibody requires further validation, and in vivo effectiveness remains to be determined. The work will be of interest to biologists working to develop pan-coronavirus vaccines and therapies.

---

## [Decision Letter]

**Decision letter after peer review:**

Thank you for submitting your article "Evaluation of the highly conserved S2 hairpin hinge as a pan-coronavirus target" for consideration by *eLife*. Your article has been reviewed by 3 peer reviewers, and the evaluation has been overseen by a Reviewing Editor and Sara Sawyer (U. Colorado Boulder) as the Senior Editor. The reviewers have opted to remain anonymous.

Essential revisions:

1) The authors' claim that the S2 hinge epitope is of interest for pan-coronavirus strategies is too speculative based on the data presented. This epitope was identified via the characterization of an antibody (3A3) isolated following immunization with a MERS S2 domain in a mouse model. Further, the fact that 3A3 binds to 2P-stabilized, but not native spike suggests that 3A3-like antibodies are unlikely to arise from natural infection. Thus, the authors should focus their key message on the discovery of a new S2 epitope and tone down the language regarding its broader implications for pan-coronavirus strategies throughout the manuscript. To more directly support the latter, the authors would need to provide evidence that 3A3-like antibodies can be isolated from human donors and/or that access to the hinge epitope can be enhanced via synergy with ACE2-mimicking antibodies or antibodies that promote S1 shedding that has been previously described.

2) Another main claim of the study is that antibody targeting the hinge is weakly neutralizing but mediates Fc-effector functions in vitro. Experiments using engineered constructs that modulate effector function (e.g., DLE (+) or LALA (-)) would more directly support this claim.

3) The chosen experimental conditions for HDXMS seemingly do not take into account the role of temperature on spike intrinsic ensemble heterogeneity (prefusion vs expanded) as previously described (PMID: 35236990). The authors must at least clarify why their HDXMS experimental conditions included thawing of frozen samples (-80 C) and brief (10 min) equilibration at 25 C instead of prolonged 37 C treatment. which favors the relevant prefusion conformation according to the above published study. Is the conformational heterogeneity in biological replicates equivalent following short incubation at 25 C and longer incubation at 37 C?

4) The S2 hinge epitope is partially occluded by the S1 domain and access is contingent on RBD up-down conformational switching. The authors should better integrate findings from the various approaches described (HDX-MS, cryo-EM, antibody engineering) to help readers understand possible mechanisms for how antibodies bind this epitope.

5) A discussion of the allosteric effects of 3A3 binding to the spike trimer, particularly with regards to cooperativity in antibody binding is warranted. In this regard, the reader would also benefit from HDXMS spectral envelopes, at least for peptides within the epitope locus.

6) In Figure 1A, regions lacking HDX-M/S coverage are indicated in grey, which is too similar to regions with coverage but with low/no rate of exchange. Please distinguish regions with and without coverage more clearly.

7) In Figure 2F, please define dashed versus solid arrows.

*Reviewer #1 (Recommendations for the authors):*

1) Readers would benefit from HDXMS spectral envelopes in figures, at least for the epitope locus peptides.

2) Authors should describe the ensemble heterogeneity observed in HDXMS and the basis for not applying a 37 deg C treatment of samples prior to HDXMS mapping studies with 3A3.

3) The reader would benefit from better integration of structural, solution (HDXMS), and antibody engineering approaches in the manuscript. This would bring out a more definitive conclusion on the mechanism of targeting the hinge epitope.

4) The authors should address the lack of allostery observed and cooperativity of antibody binding.

*Reviewer #2 (Recommendations for the authors):*

This manuscript describes work on targeting the conserved S2 hinge region, an underappreciated target that is similar to the flu stem region which universal antibodies target. Overall, this work is nicely carried out.

---

## [Author Response]

Essential revisions:1) The authors' claim that the S2 hinge epitope is of interest for pan-coronavirus strategies is too speculative based on the data presented. This epitope was identified via the characterization of an antibody (3A3) isolated following immunization with a MERS S2 domain in a mouse model. Further, the fact that 3A3 binds to 2P-stabilized, but not native spike suggests that 3A3-like antibodies are unlikely to arise from natural infection. Thus, the authors should focus their key message on the discovery of a new S2 epitope and tone down the language regarding its broader implications for pan-coronavirus strategies throughout the manuscript. To more directly support the latter, the authors would need to provide evidence that 3A3-like antibodies can be isolated from human donors and/or that access to the hinge epitope can be enhanced via synergy with ACE2-mimicking antibodies or antibodies that promote S1 shedding that has been previously described.

As suggested, we have edited several parts of the manuscript to reduce the claims around the impact of this work on pan-coronavirus strategies:

Edited title to “Identification of a conserved epitope present on spike proteins from all highly pathogenic coronaviruses” to reduce the focus on pan-coronavirus strategies.

Deleted “that can guide future pan-coronavirus strategies.” in the abstract.

Revised “Evaluation of the S2 hinge epitope… will inform design of next-generation pan-coronavirus vaccines and related therapeutic strategies” to “… will inform our understanding of the role of S2 domain epitopes in antibody recognition.”

1^st^ paragraph of Discussion section: Deleted as the focus was on S2 epitopes as pan-coronavirus targets.

Though we do not plan to isolate 3A3-like antibodies from human donors, there is evidence that these antibodies are elicited in infected humans via analysis of polyclonal responses in Claireaux *et al.* 2022. We also know of several studies on naturally occurring S2 hinge targeting antibodies from colleagues that are in preparation. Understanding the therapeutic role of this antibody class is relevant to the study of broadly-reactive S2 antibodies, even if that role is limited.

2) Another main claim of the study is that antibody targeting the hinge is weakly neutralizing but mediates Fc-effector functions in vitro. Experiments using engineered constructs that modulate effector function (e.g., DLE (+) or LALA (-)) would more directly support this claim.

We agree these are excellent controls to include, in addition to isotype controls already shown. In accordance with the *eLife* COVID research policy, we minimized our claims around Fc-effector functions elicited by RAY53 and stated that further experiments to confirm our preliminary findings are needed.

The existing description of the effector function experiments states “These results indicate that RAY53 binding is compatible with ADCP and ADCC,” which is already a very limited claim.

We also added in line 450 that S2 core-binding antibodies “require further validation” of their ability to recruit effector functions.

We appreciate the importance of controls providing effector function modulation and will include the LALAPG mutations as a standard component of our future ADCC evaluation. However, given our focus on the relevance of the epitope and consistency of the Fc regions across the antibodies, we felt that the isotype and positive control antibodies (target binding controls) were the most relevant controls to include in this study.

3) The chosen experimental conditions for HDXMS seemingly do not take into account the role of temperature on spike intrinsic ensemble heterogeneity (prefusion vs expanded) as previously described (PMID: 35236990). The authors must at least clarify why their HDXMS experimental conditions included thawing of frozen samples (-80 C) and brief (10 min) equilibration at 25 C instead of prolonged 37 C treatment. which favors the relevant prefusion conformation according to the above published study. Is the conformational heterogeneity in biological replicates equivalent following short incubation at 25 C and longer incubation at 37 C?

The HDX-MS experiments presented in this work were carried out by the D’Arcy lab and published in a preprint on bioRxiv (originally posted on February 1, 2021) prior to publication of Costello *et al.* (first posted to bioRxiv July 11, 2021, epub March 2, 2022). Indeed, our bioRxiv posting inspired the Marqusee lab to request 3A3 for inclusion in their work focused on the conformational heterogeneity of the spike protein. Without prior knowledge of the conformational heterogeneity, we carried out these epitope mapping experiments at 25 °C, which allowed us to successfully mapped the epitope without determining which conformation the antibody prefers.

The data presented in Costello *et al.* further confirms the location of 3A3’s epitope presented here and provides additional information about its preference for different conformational states within the spike protein. We have included an additional comment in the methods section stating, “The location of the 3A3 epitope was confirmed in a separate experiment carried out over the temperature range of 4 to 37 °C (Costello et al. 2022).”

This is a clear example of the value of pre-prints to stimulate timely scientific collaboration. While Costello *et al.* used 3A3 as a tool to probe spike dynamics, here we highlight the original work that identified the epitope.

4) The S2 hinge epitope is partially occluded by the S1 domain and access is contingent on RBD up-down conformational switching. The authors should better integrate findings from the various approaches described (HDX-MS, cryo-EM, antibody engineering) to help readers understand possible mechanisms for how antibodies bind this epitope.

Visualization of the 3A3/spike interaction by cryo-EM was not possible with full-length spike as the conformational heterogeneity of the S1 domains in the S2-open state was too great. For this reason, 3A3 Fab with the stabilized S2 domain of the spike were used to obtain cryo-EM images. In contrast, HDX-MS was performed with both 3A3 Fab and full-length 3A3 and full-length HexaPro spike. The accessibility of the epitope is quite different in these cases, likely resulting in different Fab occupancy depending on the presence of the S1 domains, making integration of the HDX-MS and cryo-EM results difficult.

Our main focus was to describe the location of this new class of S2 epitope, but because of the complexity of spike dynamics and the 3:2 ratio of binding sites, we lack sufficient evidence to make definitive claims about the exact mechanism of bivalent antibody binding.

We have added a new paragraph in the Discussion section to better lay out the complexity of the spike dynamics and how it may impact binding to the S2 hinge epitope:

“SARS-2 spike is a highly dynamic protein, sensitive to many variables including temperature (Edwards et al. 2021), pH (Zhou et al. 2020) and glycosylation (Casalino et al. 2020). Each of the RBDs have the potential to flip up, which has been captured in cryo-EM images of closed, one-up, two-up and three-up spikes (Benton et al. 2020). Additionally, molecular dynamics simulations indicated that movement of the S1 domains during opening extends deeper into the protein than previously appreciated (Zimmerman et al. 2021), while HDX analysis has observed an open-S2 conformation with a splayed spike trimer that exposes core S2 epitopes (Costello et al. 2022). Notably, these movements impact hinge epitope accessibility, indicating that antibody binding to this region involves multiple kinetic steps (Figure 2f). Data showing enhanced binding to spike variants favoring the S2 open state (Figure 2c-e) and similar neutralization of authentic spike on pseudo-viruses by 3A3 and RAY53 (Figure 5a) despite the latter’s increased affinity (Figure 4e) indicate that spike S2 opening is the rate-limiting step in antibody binding. Results shown here also underscore that these rates can vary depending on the local environment (soluble protein versus viral or infected cell surface) and amino acid changes present (stabilizing changes versus circulating variants), highlighting the importance of spike dynamics on antibody binding.”

5) A discussion of the allosteric effects of 3A3 binding to the spike trimer, particularly with regards to cooperativity in antibody binding is warranted. In this regard, the reader would also benefit from HDXMS spectral envelopes, at least for peptides within the epitope locus.

Thank you for this suggestion. Spectral envelopes have been provided (Supplementary Figure 4b and Supplementary Table 3).

The HDX-MS data provides limited insight into possible cooperative or allosteric binding of the 3A3 antibody because of other sources of heterogeneity such as spike dynamics and partial occupancy of the spike epitopes. However, no difference in occupancy was detected when HDX-MS with 3A3 Fab was compared to the same experiment with bivalent 3A3 IgG. It should be noted that in this HDX system, the antibody is not bound so tightly that the spectra are bimodal, showing the exchange of bound and unbound populations separately. Though HDX-MS experiments were performed in slight Fab or IgG excess of 1:1 Fab:spike monomer stoichiometry, the absolute stoichiometry in the context of the spike trimer is unclear.

6) In Figure 1A, regions lacking HDX-M/S coverage are indicated in grey, which is too similar to regions with coverage but with low/no rate of exchange. Please distinguish regions with and without coverage more clearly.

Thank you for this suggestion, this has been edited so that no coverage is black instead of grey in both Figure 1a and Supplementary Figure 4a.

7) In Figure 2F, please define dashed versus solid arrows.

Thank you, we have edited this image to include only solid arrows.

Reviewer #1 (Recommendations for the authors):1) Readers would benefit from HDXMS spectral envelopes in figures, at least for the epitope locus peptides.2) Authors should describe the ensemble heterogeneity observed in HDXMS and the basis for not applying a 37 deg C treatment of samples prior to HDXMS mapping studies with 3A3.3) The reader would benefit from better integration of structural, solution (HDXMS), and antibody engineering approaches in the manuscript. This would bring out a more definitive conclusion on the mechanism of targeting the hinge epitope.4) The authors should address the lack of allostery observed and cooperativity of antibody binding.

Please see the response to Essential Revisions #3, 4, and 5 discussing these points.